# Review: The Calibration of DEM Parameters for the Bulk Modelling of Cohesive Materials

Corné J. Coetzee [†,*] and Otto C. Scheffler [†]

Department of Mechanical & Mechatronic Engineering, Stellenbosch University, Stellenbosch 7600, South Africa
* Correspondence: ccoetzee@sun.ac.za; Tel.: +27-21-808-4239
† These authors contributed equally to this work.

**Abstract:** Granular materials are abundant in nature, and in most industries, either the initial constituents or final products are in granular form during a production or processing stage. Industrial processes and equipment for the handling of bulk solids can only be improved if we can understand, model and predict the material behaviour. The discrete element method (DEM) is a numerical tool well-suited for this purpose and has been used by researchers and engineers to analyse various industrial applications and processes. However, before any bulk scale modelling can be undertaken, the input parameters must be carefully calibrated to obtain accurate results. The calibration of parameter values for non-cohesive materials has reached a level of maturity; however, the calibration of cohesive materials requires more research. This paper details the most prevalent contact models used to model cohesive materials—presented in a consistent notation. Moreover, the significant differences between the models are highlighted to provide a reference for engineers and researchers to select the most appropriate model for a specific application. Finally, a critical review of calibration experiments and methodologies often used for cohesive materials is also presented. This provides a solid basis for DEM practitioners to select the most appropriate calibration methodology for their application and for researchers to extend the current state-of-the-art practices.

**Keywords:** cohesive material; discrete element method; contact models; bulk calibration; review





## 1. Introduction

The behaviour of granular materials is complex, exhibiting both solid, fluid and gas-like properties. However, our understanding and ability to accurately model and predict the behaviour of these materials is limited and even more so when the material is cohesive.

The handling of bulk solids in granular form is found in many industries, such as the mining, agriculture, pharmaceutical, and food industries. When, for example, mining under the water table or during the rainy season, the run-of-mine material is wet and can contain high levels of clay and shale, rendering it extremely cohesive. Cohesive materials tend to build up on structures and equipment, leading to arching, blocking and operating downtime.

The discrete element method (DEM) is a numerical tool for modelling granular materials and predicting their behaviour. In DEM, a finite number of individual or discrete particles are modelled, and the contact model and particle shape and size distributions govern the bulk behaviour. The contact models for non-cohesive materials consist of linear or non-linear springs, dashpots (damping) and frictional sliders (translation/sliding and rolling).

However, when the material is wet (moist), capillary forces due to liquid bridges at the contacts result in bulk cohesive behaviour. Therefore, cohesive elements should be included in the contact model to account for this behaviour. Several such models have been developed, the most prevalent of which are presented in the first part of this paper.

Although the contact models have been presented elsewhere, we provide a uniform notation and set of symbols, making it easier for novice users to comprehend. Some of

the significant differences between the models are also highlighted. We also propose a new naming convention for the simplified Johnson-Kendall-Roberts (*simplified* JKR or SJKR) models. Various software codes (open-source, commercial and academic) have implemented these models using different names, such as SJKR, SJKR1 or SJKR2, without providing a specific formulation and leaving the reader uncertain. In some cases, the same name is used for slightly different implementations. Although the models are all related and can provide the equivalent solution by simply scaling the value of the cohesive parameter (cohesion energy density), the parameter values used are less valuable to other users and researchers if the specific implementation is not provided. Therefore, instead of using a numerical naming convention, we propose an alphabetic convention (i.e., SKR-A, SJKR-B, etcetera) which, if adopted, will provide a clear definition for future reference.

Due to several simplifications and assumptions, the input parameter values (mostly contact parameters) need to be calibrated before the model can accurately predict the bulk behaviour of a specific material. Assumptions include the modelling of contact physics and the particle shape and size distributions. Hence, measuring the particle and contact properties at the micro or meso scale will not necessarily provide accurate predictions at the bulk (macro) scale. Therefore, a systematic calibration process is needed to compensate for the simplifications and assumptions. For example, due to computational limits, we cannot include the surface texture of the particles in our model, nor can we accurately model the shape and size of every individual particle. However, the bulk shear resistance of the material, as measured in a direct shear test, for example, is a combination of the particle-particle coefficient of friction (surface texture) and the mechanical interlocking of the particles due to their shape. Thus, we need to select a simplified particle shape and a feasible size distribution and then adjust the coefficient of friction to compensate for these simplifications.

The calibration of a DEM model for non-cohesive materials has been studied by several authors [1–13], as summarised and reviewed by Coetzee [14], who introduced two philosophies. In the *Direct Measuring Approach*, the material properties are measured at the particle and contact level and directly used in the model. However, as described above, this approach cannot compensate for the simplifications and assumptions made in the physics and implementation. On the other hand, in the *Bulk Calibration Approach*, laboratory or field tests are conducted, and one or more bulk material property measured (e.g., angle of repose). The experiment is then repeated numerically, and the input parameters are adjusted until the same bulk response is predicted. Here it is essential to select an experiment sensitive to the input parameters. Also, using only a single experiment to calibrate more than one input parameter can be troublesome since a unique set of parameter values cannot be guaranteed. Therefore, combining more than one experiment is advised to obtain a unique set of parameter values [8,13].

Compared to non-cohesive materials, the contact models for cohesive materials have at least one additional input parameter that requires calibration. Furthermore, regarding bulk behaviour, the cohesive parameter usually does not act independently, and it is the combined effect of this parameter and all the 'non-cohesive' parameters that should be analysed and calibrated. As such, when calibrating cohesive materials, more simulation runs are needed compared to non-cohesive materials.

Experienced users can simply iteratively change the parameter values and manually converge to a calibrated parameter set. However, an optimisation approach can effectively calibrate many input parameters, as summarised and reviewed by Richter et al. [15]. Nevertheless, even when using an optimisation scheme to find a calibrated set of parameter values, the experiments should still be sensitive to changes in the material properties and the model input parameters. For example, if the angle of repose is measured, it should be sensitive to changes in the material's bulk cohesion—if the moisture content is increased (statistically different), we should see a statistically different angle of repose. On the other hand, if the angle of repose does not change, then the experiment is not sensitive enough to changes in material cohesion (in the required range), and a different experiment should be

used. Similarly, the model should also be sensitive to changes in the input parameters— if the contact cohesion parameter is changed, the modelled bulk behaviour should change accordingly. If this is not the case, an optimisation algorithm will still provide a calibrated or 'optimised' set of parameter values, but it might not be accurate in any other application or experiment. Thus, before any optimisation technique can be applied to a calibration methodology, we first need to ensure that both the physical and numerical experiments are sensitive to material properties and input parameter changes, respectively.

Therefore, the second part of this paper strives to identify the experiments and methodologies successfully utilised to calibrate the input parameters of cohesive materials. Note that the scope is limited mainly to mining and, to some degree, the agricultural sectors. The focus is not on the pharmaceutical and food industries, where cohesive powders are often encountered.

This paper is formulated as follows: Section 2 provides the reader with a brief overview of the forces responsible for material cohesion, Section 3 provides an overview and detailed formulation of the most prevalent cohesion contact models, Section 4 critically reviews the experiments, contact models and methodologies used in calibrating the input parameters for various cohesive materials, Section 5 compares the computational performance of the different contact models as reported in the literature, and lastly, Section 6 concludes and provides an outlook for future research.

In physics, the term *cohesion* is customarily reserved for the sticking together of particles of the same substance, while the term *adhesion* is used for different substances. However, in the field of bulk material handling and DEM modelling, these two terms are used interchangeably. Some contact models are referred to as *cohesion models*, while others are referred to as *adhesion models* without any clear distinction.

## 2. Cohesive Forces

A brief overview of the electrostatic, van der Waals and capillary/liquid-bridge (both terms are used interchangeably in literature) forces is presented. Furthermore, their significance, relating to particle size, is compared when modelling a bulk system. For a detailed review of their mechanics, the reader is referred to Visser [16] and Seville et al. [17].

### 2.1. Electrostatic Forces

DEM can model the behaviour of particles where the cohesive effects arising from electrostatic forces may be of interest. Electrostatic forces occur between particles due to electric charges on the particles' surface. Coulomb's law describes the electrostatic force between two spherical particles [16]. Including additional body forces in DEM is straightforward and can readily be incorporated into a contact model. However, a significant difficulty arises when attempting to quantify the electric charges on the surface of granules—even approximately [18]. Nevertheless, Duran [18] does describe an empirical estimation technique for assessing the surface charge.

Electrostatic forces will only be significant for dry systems, as moist systems (with an electrolyte present) will discharge. Moreover, the electrostatic discharge will even occur in humid environments [16]. Additionally, electrostatic forces are far weaker than van der Waals and capillary forces (as illustrated in Figure 1), with these forces only becoming significant at particle sizes of ($<10$ μm) [19]. Furthermore, van der Waals forces will also become significant at this length scale. Attempting to simulate a system on this scale will be highly intensive, and separately modelling individual cohesive phenomena will be prohibitively demanding. Consequently, incorporating electrostatic forces for large bulk systems is negligible—especially if moisture is present in the system. For a detailed review of electrostatic forces in granular flow, see Zhao et al. [20].

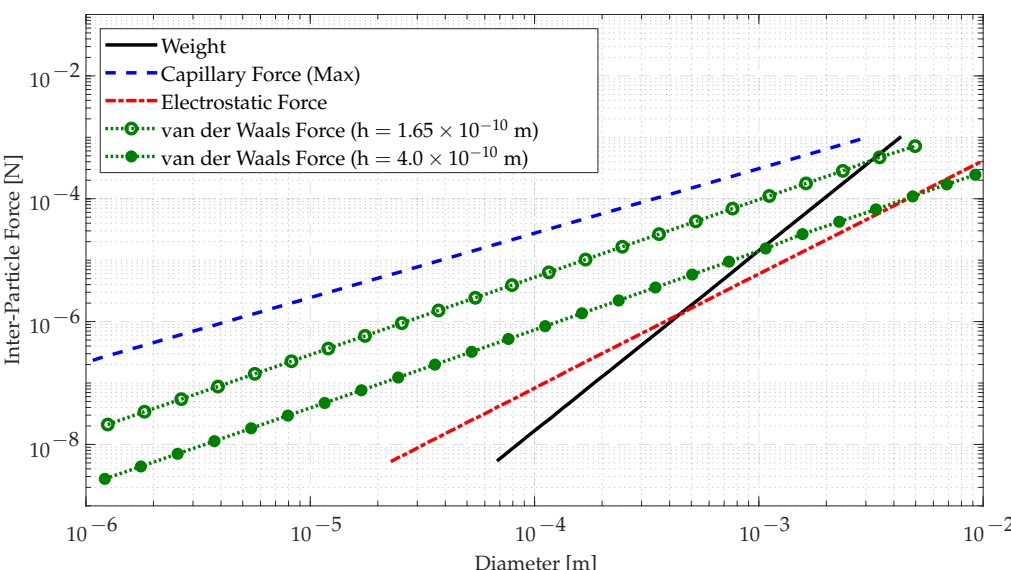

**Figure 1.** Comparison of theoretical inter-particle force magnitudes (adapted from [17]). Reproduced with permission from Powder Technology; published by Elsevier, 2000.

## 2.2. Van der Waals Forces

Van der Waals Forces are approximately 50 times larger than electrostatic forces [21]. In contact mechanics, the term 'van der Waals forces' collectively describes the entirety of the intermolecular forces which always exist between atoms and molecules [22]. Subsequently, these forces give rise to surface adhesion/cohesion between macroscopic bodies. Accordingly, the total van der Waals force on a macroscopic body can be determined by integrating over the total volume of the object.

Van der Waals forces are a maximum when the separation distance between the bodies takes a minimum value in the order of the intermolecular spacing [17]. However, no material is perfectly rigid; therefore, the contact force between objects is distributed over a finite contact area [23]. This occurrence gave rise to Hertz's description of contact behaviour. In the absence of electrostatic and capillary forces, the van der Waals adhesive force acting inside and on the periphery of this contact area (alongside the elastic Hertzian contact) is described by the well-known Johnson, Kendall & Roberts (JKR) theory [22] and is detailed in Section 3.4.

## 2.3. Liquid-Bridge Forces

The liquid bridge is principally responsible for the cohesion of granular materials in a pendular state. The mechanical properties of a system of wet granular material will alter depending on the amount of moisture present. Researchers typically describe the various degrees of liquid present in a system as the liquid saturation $S$, a ratio of the liquid volume $V_L$ and void volume $V_V$ of the system. Accordingly, the saturation can be computed from the volume fractions of the liquid $\Phi_L$ and solids $\Phi_S$ in the system and is described by,

$$S = \frac{V_L}{V_V} = \frac{\Phi_L}{(1 - \Phi_S)} \tag{1}$$

Five different saturation states have been defined depending on the quantity of moisture in the system. An overview of the different states is given by Mitarai and Nori [24] in Table 1, who reviews Iveson et al. [19] and Newitt and Conway-Jones [25].

**Table 1.** States of saturation [24].

| Liquid Content | State | Schematic Diagram | Physical Description |
|---|---|---|---|
| None | Dry | | Cohesion is negligible. |
| Minimal | Pendular | | Cohesion acts through liquid-bridges. |
| Transitional | Fenicular | | Cohesion acts through liquid-bridges and liquid filled pores. |
| Saturated | Capillary | | All voids are filled with liquid. The surface liquid is drawn into the pores due to capillary action, which gives rise to particle cohesion. |
| Over Saturated | Droplet/Slurry | | No cohesion occurs as liquid pressure is equal to or greater than that of the air. |

In a dry state, there is no moisture present, and cohesion is negligible. This is contrasted with a pendular state, where low moisture levels are present, but saturation levels remain below 25% [26]. In a funicular state, fluid webs are interspersed with air pockets. In a capillary state, all voids are filled with fluid, but menisci can still form at interfaces between fluid and air, which allows capillary action to take place.

However, note that the pendular assumption that *independent* liquid-bridges will occur for low saturation levels relies on the microstructural arrangement and porosity of the specific system [27]. As such, researchers have studied systems with saturation levels slightly lower than 20% [28], with Scholtès et al. [27] finding that menisci are only strictly independent for $S < 12\%$, while insignificant menisci overlap occurs until $S < 15\%$. The expressions governing the force arising from the liquid bridge are further detailed in Section 3.6.

*2.4. Cohesive Force Comparison*

Seville et al. [17] compared the inter-particle forces (due to the electrostatic, van der Waals and capillary forces) with particle weight for quasi-static systems. Suitable values for the various variables and a particle density of $1000 \, \text{kg/m}^3$ were specified—resulting in Figure 1. Subsequently, only particles less than 0.5 mm in diameter may experience an electrostatic force exceeding their respective particle weight. Similarly, van der Waals forces exceeding their particle weight are also restricted. The latter occurs for spherical particle diameters of 1 to 3 mm.

Furthermore, the capillary force may exceed the particle weight for diameters up to 5 mm. However, it should be noted that this represents ideal spheres, and if surface roughness is accounted for (affecting the separation distance h between the particles), these diameters may be much less. For example, the van der Waals forces will only become significant for particle diameters of <100 µm [17].

For dynamic situations, Herminghaus [29] demonstrated that grains with diameters of ≈40 µm and velocities of ≈0.01 m/s would result in kinetic energies of roughly $5 \times 10^{-16}$ J. This energy is significantly larger than the van der Waals binding energy. Accordingly, van der Waals forces can be disregarded for the particle sizes typically encountered in granular flows but may be significant in powder flows [16]. Consequently, electrostatic and van

der Waals forces can be safely neglected when the liquid-bridge forces are present and dominate the cohesive effects.

## 3. Contact Models for the Modelling of Cohesive Materials

The most prevalent contact models for the modelling of cohesive materials are presented. It is assumed that the reader is familiar with DEM, contact models for non-cohesive materials and their parameters (a comprehensive review of the normal and tangential contact models is given by Kruggel-Emden et al. [30] and Kruggel-Emden et al. [31], respectively).

The main difference between contact models for cohesive and non-cohesive materials is the formulation of the normal force component. Although the tangential (shear) force model might also be different (see Section 3.8), the normal force is primarily responsible for the bulk cohesive behaviour and is, therefore, the focus of this review. The viscous damping force, which acts in parallel with the normal (and tangential) model, is in most cases identical for cohesive and non-cohesive models and thus not presented here (an exception might be the liquid-bridge model where the damping is due to liquid viscosity, [32]).

The following sign convention is used throughout the paper: positive contact forces push the pieces apart, and negative forces pull them closer together. Positive displacement increments occur when two pieces move closer together, and when they numerically intersect, a positive overlap occurs. Negative overlaps occur when there is a separation distance or gap between the two pieces.

### 3.1. Rolling Resistance Contact Model

Rolling resistance is frequently utilised when spherical particles are employed in simulations. Spherical particles facilitate easy and efficient contact detection without requiring particle orientation tracking [14]. Furthermore, the original development of Cundall and Strack [33] used disks (for the two-dimensional application) as many discrete particles can be simulated without excessive memory requirements. Subsequently, initial advances in DEM focused on developing fast and robust algorithms for spherical particle applications. Unfortunately, physical particles are never perfectly spherical—unlike their numerical counterparts.

All physical particles will exhibit slight surface adhesion, surface roughness and a finite contact area due to minute deformations – the latter being the basis of Hertz's contact theory. As such, an artificial property known as *rolling resistance* has been introduced, which applies a moment that opposes relative particle rotation. This moment compensates for the numerical model's over-simplistic particle representation when spherical particles or other simplified shapes are employed.

In theory, a rolling resistance model can be implemented in parallel with any non-cohesive or cohesive contact force model. Similar to normal and tangential contact force models, several rolling resistance models are available. The type C model, as designated by Ai et al. [34], is presented here. This model is also recommended by Wensrich and Katterfeld [35] and is presented according to its implementation in PFC [36] (additionally, the reader is referred to Ai et al. [34] for a comprehensive review of various other rolling resistance models).

The rolling resistance moment $\boldsymbol{M_r}$ is incrementally updated,

$$\boldsymbol{M_r^*} = \boldsymbol{M_r^t} + k_r \boldsymbol{\Delta\theta_b} \tag{2}$$

where $t$ is the time-step, $k_r$ is the rolling resistance stiffness, and $\boldsymbol{\Delta\theta_b}$ is the increment in bend-rotation (see [36] for definition). The rolling stiffness is given by,

$$k_r = k_s (R^*)^2 \tag{3}$$

where $k_s$ is the shear stiffness (in the tangential direction) and $R^*$ the effective contact radius given in terms of the particle radii $R_{1,2}$:

$$R^* = \frac{R_1 R_2}{R_1 + R_2} \tag{4}$$

The updated moment is then checked against the limit value,

$$M_r^{t+1} = \begin{cases} M_r^* & \text{if } | M_r^* | \leq \mu_r R^* \left( F_n - F_{po} \right) \\ \mu_r R^* \left( F_n - F_{po} \right) \left( \frac{M_r^*}{|M_r^*|} \right) & \text{otherwise} \end{cases} \tag{5}$$

where the normal force, $F_n$, is calculated based on the specific contact model used, and $\mu_r$ is the coefficient of rolling friction.

Furthermore, the Coulomb friction limit is adjusted to account for cohesive effects in the tangential direction. The latter is readily implemented and based on the work of Thornton [37] and Thornton and Yin [38], who showed that with adhesion present, the contact area decreases with an increase in the tangential force. That is, a transition from a 'peeling' action to a 'sliding' action occurs as the tangential force reaches a critical value. At this critical point, the contact area corresponds to a Hertzian-like normal stress distribution under a load of $\left( F_n^H - F_{po} \right)$, assuming the JKR theory (Section 3.4 [39]), where $F_n^H$ is the elastic force according to Hertz's theory and $F_{po}$ is the pull-off (maximum tensile/adhesive) force [40–42].

The same principle is applied to the rolling resistance model where the maximum tensile (cohesive) force, $F_{po}$, is added to the normal force $F_n$ (Equation (5)) to increase the reference force. Note that $F_{po}$ will always have a negative sign; thus, it is subtracted from the always positive $F_n$ to increase the reference force.

### 3.2. Luding's Contact Model

Materials subjected to relatively high loads (consolidation) may experience plastic deformation, resulting in increased contact areas and increased cohesive behaviour. History-dependent elasto-plastic-adhesive contact models can be used in DEM to simulate this behaviour. One such model was developed by Luding [43] and improved gradually over time (see [43–46]). The model is an extension of the linear hysteretic model developed by Walton and Braun [47]. Luding's model expanded the original development by allowing tensile forces to develop, as shown in Figure 2, and is subsequently conveyed according to Coetzee's [48] implementation. The load-displacement curve is generally defined by the constant pull-off force, $F_0$, the loading branch stiffness $k_1$, the unloading/reloading branch stiffness $k_2$, the adhesion branch stiffness $k_a$, and the plastic overlap (deformation) $\delta_p$. The force is given by,

$$F_n^L = \begin{cases} F_0 + k_1 \delta_n & \text{if } k_2(\delta_n - \delta_p) \geq k_1 \delta_n \\ F_0 + k_2(\delta_n - \delta_p) & \text{if } k_1 \delta_n > k_2(\delta_n - \delta_p) > -k_a \delta_n \\ F_0 - k_a \delta_n & \text{if } -k_a \delta_n \geq k_2(\delta_n - \delta_p) \end{cases} \tag{6}$$

Contact is activated at a normal overlap $\delta_n = 0$, where the force jumps to a value $F_0$ (which should take a negative value). Virgin loading follows the $k_1$-branch defined by the stiffness $k_1$ (the first condition in Equation (6)). The maximum overlap $\delta_{max}$ is a history-dependent parameter and is updated and stored with the contact.

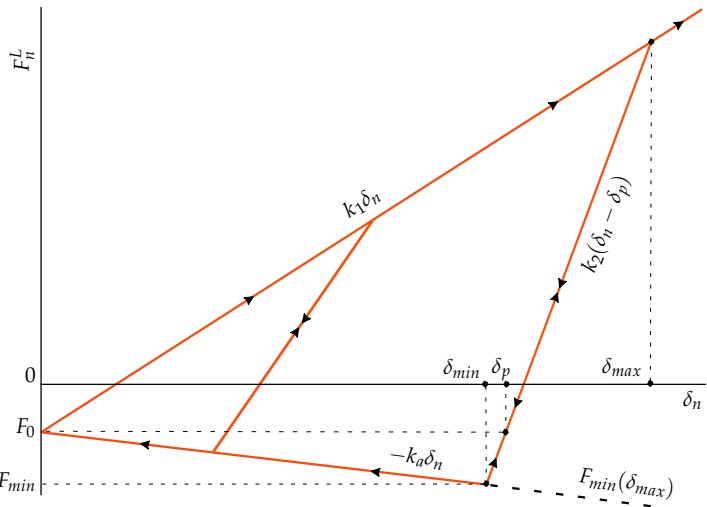

**Figure 2.** Luding's model force-displacement relation in the normal direction.

Unloading follows the $k_2$-branch defined by the stiffness $k_2$ and the plastic overlap $\delta_p$ (the second condition in Equation (6)). The plastic overlap is defined as the overlap where the force on the $k_2$-branch is equal to the pull-off force $F_0$ and is given in terms of the maximum overlap,

$$\delta_p = \left(1 - \frac{k_1}{k_2}\right)\delta_{max} \tag{7}$$

Unloading along the $k_2$-branch results in a tensile (adhesive) force, and at the plastic overlap $\delta_p$, the force is equal to the pull-off force $F_0$. Further unloading follows the same branch until the adhesion branch is met, where the minimum force (maximum adhesion) is given by,

$$F_{min} = F_0 - k_a\delta_{min} \tag{8}$$

where the overlap at the minimum force is given by:

$$\delta_{min} = \left(\frac{k_2 - k_1}{k_2 + k_a}\right)\delta_{max} \tag{9}$$

Once the adhesion branch is met during unloading on the $k_2$-branch, further unloading switches to the adhesion branch (the third condition in Equation (6)). If the contact is unloading along the $k_2$-branch and then starts to reload, the same $k_2$-branch is followed until the $k_1$-branch is reached. Further virgin loading follows the $k_1$-branch and will result in an increased and updated $\delta_{max}$. If the contact is unloading along the $k_a$-branch and then starts to reload, a $k_2$-branch is followed, as shown in Figure 2. In this case, the value of $\delta_p$ is updated to where this $k_2$-branch is equal to the pull-off force $F_0$.

Luding [43] acknowledges that the unloading/reloading path should probably be non-linear to be more realistic and to account for the larger contact area (higher stiffness) with an increase in plastic deformation. However, due to a lack of detailed experimental information, the piece-wise linear implementation is used, and a refinement is added to increase the unloading/reloading stiffness $k_2$ as a function of maximum overlap,

$$k_2 = \begin{cases} k_2^{max} & \text{if} \quad \delta_{max} \geq \delta_{max}^* \\ k_1 + (k_2^{max} - k_1)\frac{\delta_{max}}{\delta_{max}^*} & \text{if} \quad \delta_{max} < \delta_{max}^* \end{cases} \tag{10}$$

where $k_2$ increases from $k_1$ to the maximum allowable value $k_2^{max}$ as specified by the user, and the plastic flow overlap limit is given in terms of the harmonic mean radius $R_h$,

$$\delta_{max}^* = \frac{k_2^{max}}{k_2^{max} - k_1} \phi_f R_h \tag{11}$$

where $\phi_f$ is the user-defined dimensionless plasticity depth and,

$$R_h = \frac{2R_1 R_2}{R_1 + R_2} \tag{12}$$

If the one piece is a wall, $R_h = 2R_p$, where $R_p$ is the particle radius. With $\phi_f = 0$, $\delta_{max}^* = 0$, and the first condition in Equation (10) will always be met with $k_2 = k_2^{max}$. Furthermore, the stiffness $k_2^{max}$ is defined in terms of the plasticity ratio $\lambda_p$,

$$k_2^{max} = \frac{k_1}{1 - \lambda_p} \quad \text{or} \quad \lambda_p = 1 - \frac{k_1}{k_2^{max}} \tag{13}$$

when $\lambda_p = 0$, $k_2^{max} = k_1$ and there is no plastic behaviour, and when $\lambda_p \to 1$, $k_2 \to \infty$ and the behaviour is perfectly plastic. The adhesion stiffness $k_a$ is also defined relative to the $k_1$ stiffness using the factor $k_{af}$,

$$k_a = k_{af} k_1 \tag{14}$$

### 3.3. Edinburgh Elasto-Plastic Adhesion Model

The Edinburgh Elasto-Plastic Adhesion Model (EEPA) is a progression of the adhesive hysteretic spring model presented by Luding [43] (Section 3.2). This model was further developed by Morrissey [49] and implemented in EDEM [50]. The model is especially suited for simulating materials subjected to relatively high loads (consolidations) that may result in plastic deformation at the contact points. Accordingly, increased contact areas may manifest at the contact points, which results in increased cohesive behaviour. That is, the plastic deformation at the contact points gives rise to stress-history-dependent behaviour. Therefore, the EEPA model is advantageous when modelling systems where these effects significantly influence the bulk response of the system.

The load-displacement curve is given in Figure 3, and the formulation of the normal force is given by,

$$F_n^{EEPA} = \begin{cases} F_0 + k_1 \delta_n^m & \text{if} & k_2(\delta_n^m - \delta_p^m) \geq k_1 \delta_n^m \\ F_0 + k_2(\delta_n^m - \delta_p^m) & \text{if} & k_1 \delta_n^m > k_2(\delta_n^m - \delta_p^m) > -k_a \delta_n^{\chi} \\ F_0 - k_a \delta_n^{\chi} & \text{if} & -k_a \delta_n^{\chi} \geq k_2(\delta_n^m - \delta_p^m) \end{cases}$$

with: $k_1$ – Effective initial contact loading stiffness $[\text{N/m}]$

$k_2$ – Effective unloading/reloading contact stiffness $[\text{N/m}]$

$k_a$ – Adhesive unloading contact stiffness $[\text{N/m}]$

$\delta_p$ – Plastic contact deformation $[\text{m}]$

$F_0$ – Constant pull-off force $[\text{N}]$

$m$ – Loading/unloading/reloading exponent $[-]$

$\chi$ – Adhesive branch unloading exponent $[-]$

$$\tag{15}$$

Initial contact occurs at zero overlap, when the force jumps to the constant pull-off force $F_0$. Hereafter, the force follows the *initial loading* branch until the history-dependent maximum overlap $\delta_{max}$ is reached. After that, the unloading occurs along the so-called *unloading/reloading* branch. Continued unloading will eventually result in a tensile force, and the plastic deformation $\delta_p$ is the point where this force is equal to $F_0$. Further unloading will result in the maximum tensile force $F_{min}$ being reached where the normal overlap is dubbed $\delta_{min}$, given by,

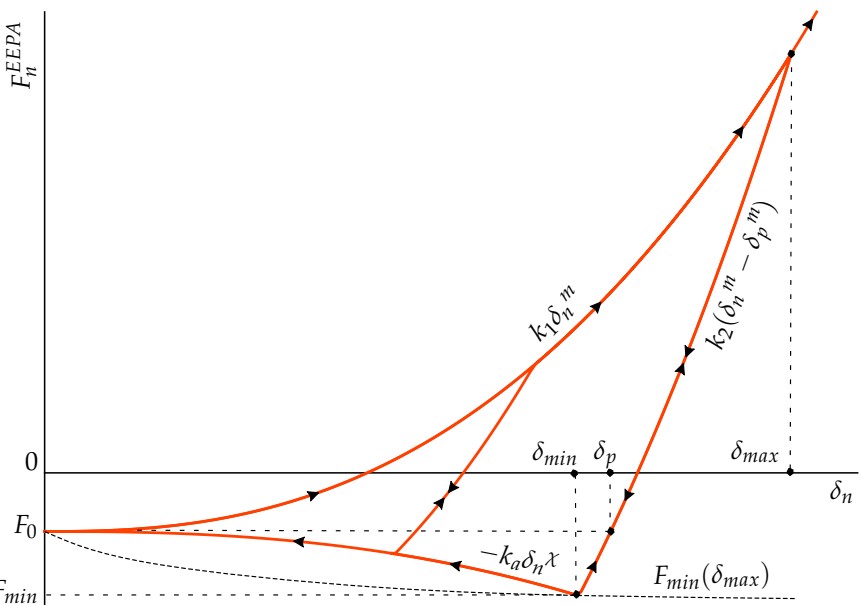

**Figure 3.** EEPA model force-displacement relation in the normal direction.

$$\delta_{min} = \left( \delta_p^m + \frac{F_{min} - F_0}{k_2} \right)^{\frac{1}{m}} \tag{16}$$

Hereafter, subsequent unloading switches to the *adhesive* branch until zero overlap is reached and the force is equal to $F_0$. Accordingly, the adhesive stiffness is given by,

$$k_a = \frac{F_{min} - F_0}{\delta_{min}^{\chi}} \tag{17}$$

The incorporation of an *initial loading* and *unloading/reloading* branch exponent ($m$) introduces non-linearity. Additionally, the deformation exponent $\chi$ introduces non-linear softening behaviour to the *adhesive* branch [51] with a sudden decrease in adhesive strength, which is better aligned with the microscopic observations of Jones [52]. The relationship between the *initial loading* and *unloading/reloading* stiffnesses is governed by the contact plasticity ratio $\lambda_p$,

$$\lambda_p = 1 - \frac{k_1}{k_2} \quad \text{or} \quad k_2 = \frac{k_1}{1 - \lambda_p} \tag{18}$$

When $\lambda_p = 0$, then $k_2 = k_1$, and no plastic contact deformation occurs, whilst $\lambda_p \to 1$ results in $k_2 \to \infty$, and the contact deformation behaviour is perfectly plastic. The plastic contact deformation/overlap corresponds to the overlap at $F_0$ and may be determined through utilisation of the maximum overlap and plasticity ratio as,

$$\delta_p = \delta_{max} \lambda_p^{1/m} \tag{19}$$

The specification of $F_{min}$ differs from Ludings's formulation, with the modeller indirectly controlling this force. This is performed by specifying a surface energy $\gamma^*$ to determine the maximum attractive tensile force [49]. Thus, $F_{min}$ closely resembles the JKR pull-off force (Section 3.4),

$$F_{min} = F_0 - \frac{3}{2}\pi\gamma^* a \tag{20}$$

The contact patch radius $a$ is determined at the point of plastic overlap (deformation) $\delta_p$ based on Hertz's contact theory,

$$a = \sqrt{2\delta_p R^*} \tag{21}$$

However, the implementation by EDEM [50] utilises the geometric lens area between two overlapping spheres,

$$a = \sqrt{\frac{\pi}{4d^2}\left(4d^2 R_1^2 - \left(d^2 - R_2^2 + R_1^2\right)^2\right)}$$

(22)

with: $d$ – Centre distance at the plastic overlap between particles [m]

Analysis shows that the radius calculated by Equation (22) is approximately equal to that of Equation (21). That is, the difference is less than 2% when the overlap ranges from 0 to 5% of the effective particle radius $R^*$, whilst the ratio of the two-particle radii $R_2/R_1$ ranges from 1 to 10. Consequently, Equation (21)'s more efficient formulation of the contact area is preferred.

The model by Luding (Section 3.2) is very similar to the EEPA model. However, where the EEPA model has a linear (by setting the exponents $m = \chi = 1$) and non-linear option, Luding's model assumes only linear load-displacement relations for all loading-unloading branches. One of the differences, however, is the specification of the minimum force (maximum adhesion). In the EEPA model, the minimum force is set by specifying the surface adhesion energy based on the Johnson-Kendall-Roberts (JKR) model [39]. Unloading continues along the $k_2$-branch until the minimum force is reached. Any further unloading follows the adhesion branch, where the adhesion stiffness, $k_a$, is calculated based on the contact history to ensure that the force returns to the pull-off force, $F_0$, when $\delta_n = 0$. In Luding's model, however, the minimum force is not explicitly set, but rather the adhesion stiffness $k_a$. Unloading along the $k_2$-branch continues until the adhesion branch is reached, and any further unloading will then follow the adhesion branch. Furthermore, Luding's model has the option of having the $k_2$ stiffness history-dependent; this behaviour is not available in the EEPA model.

In summary, the six input parameters needed for the EEPA model are the initial loading stiffness $k_1$, the plasticity ratio $\lambda_p$, the *loading/unloading* displacement exponent $m$, the *adhesive* displacement exponent $\chi$, the constant pull-off force $F_0$ and the surface (adhesion) energy density $\gamma^*$.

### 3.4. Johnson, Kendall and Roberts (JKR) Model

The JKR theory was defined by Johnson et al. [39] and incorporated attractive surface energy into a modified Hertz formulation to account for the attractive surface forces, $F_n^{JKR}$, originating from van der Waals effects. Subsequently, a larger contact area—than that predicted by Hertz's theory—is formed due to the model's attractive component. However, the theory has also been utilised to model materials where the adhesion is caused by capillary or liquid-bridge forces [42,49,53,54]. The JKR description of the enlarged contact area's radius is subsequently obtained by solving for the single real root of Equation (23),

$$a^3 = \frac{3R^*}{4E^*}\left[F_n^{JKR} + 3\pi\gamma^* R^* + \sqrt{F_n^{JKR} 6\pi\gamma^* R^* + (3\pi\gamma^* R^*)^2}\right]$$

with: $E^* = \left(\frac{(1-\nu_1^2)}{E_1} + \frac{(1-\nu_2^2)}{E_2}\right)^{-1}$ – Effective elastic modulus,

$E_1$ & $E_2$ [Pa] are the elastic moduli of the contacting bodies and,

(23)

$\nu_1$ & $\nu_2$ are the Poisson's ratios [–]

$\gamma^* = \gamma_1 + \gamma_2 - \gamma_{12}$ – Work of adhesion,

$\gamma_1$ & $\gamma_2$ are the intrinsic surface energies of two contacting bodies and

$\gamma_{12}$ is the interface energy $\left[\mathrm{J/m^2}\right]$

If both surface energies are equal, then $\gamma^*$ becomes $2\gamma$ (i.e., the total adhesion energy of both surfaces). To determine van der Waals interactions, the approximating combining law $\gamma^* = \sqrt{\gamma_1 \times \gamma_2}$, may be used [55]. The normal contact force between the spheres may then be calculated as,

$$F_n^{JKR} = \frac{4E^*a^3}{3R^*} - \sqrt{8\pi\gamma^*E^*a^3} \tag{24}$$

where the first term is exactly that of Hertz's contact theory. The radius $a_0$ of the contact area where the force $F_n^{JKR}$ is zero (equilibrium point when all externally applied forces are zero) can be found as,

$$a_0 = \left(\frac{9\pi\gamma^*R^{*2}}{2E^*}\right)^{1/3} \tag{25}$$

From Equation (23), it is observed that when $F_n^{JKR}$ becomes negative, a real solution will only be obtained when,

$$(3\pi\gamma^*R^*)^2 \geq 6\pi F_n^{JKR}\gamma^*R^* \tag{26}$$

Accordingly, the maximum tensile force occurs when the so-called pull-off force $F_{po}$ is reached,

$$F_{po} = -\frac{3\pi\gamma^*R^*}{2} \tag{27}$$

The contact radius $a_{po}$ corresponding to the maximum pull-off force can be found utilising Equations (24) and (27),

$$a_{po} = \left(\frac{9\pi\gamma^*R^{*2}}{8E^*}\right)^{1/3} \tag{28}$$

Hereafter, the contact radius can be related to the normal displacement $\delta_n$ as specified by Johnson [56]:

$$\delta_n = \frac{a^2}{R^*} - \left(\frac{2\pi\gamma^*a}{E^*}\right)^{1/2} \tag{29}$$

The contact radius will decrease as the particles separate, with the contact radius $a_0$ corresponding to the transition point of compression to tension. As the tension force increases in magnitude (i.e., becoming more negative), the two particles stretch with a *neck* forming between the two. The rupture of this *neck* coincides with a tear-off force $F_{to}$ and corresponding tear-off contact radius $a_{to}$ specified by Johnson [56] as,

$$F_{to} = \frac{5F_{po}}{9} \quad \text{and} \quad a_{to} = \frac{a_{po}}{3^{2/3}}$$

$$\therefore a_{to} = \left(\frac{\pi\gamma^*R^{*2}}{8E^*}\right)^{1/3} \tag{30}$$

Hence, the negative displacement at tear-off $\delta_{to}$ can be found as,

$$\delta_{to} = -\frac{1}{2}\frac{1}{6^{1/3}}\frac{a_0^2}{R^*} \tag{31}$$

The JKR model allows the contact interaction to be described entirely with knowledge only from the physical properties of the materials in question. The model's behaviour is shown in Figure 4, utilising dimensionless relations [55]. In addition, illustrations of the collision profile at critical points are provided for further clarification.

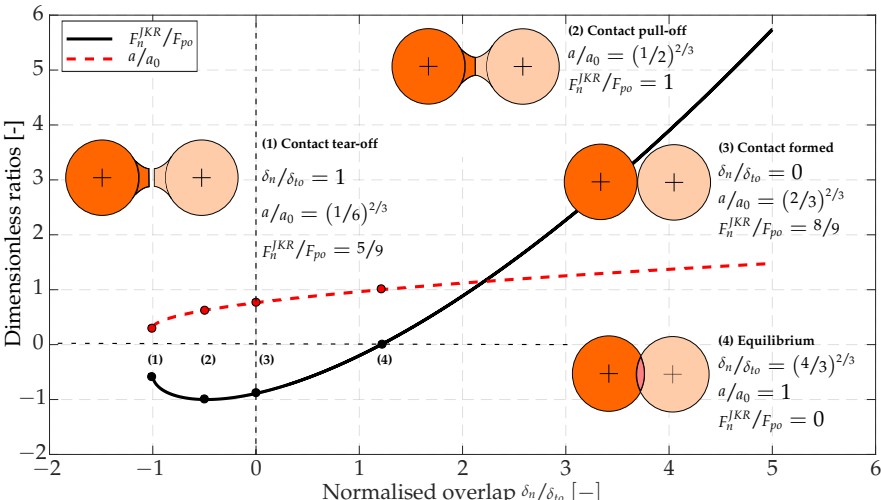

**Figure 4.** Schematic example of the JKR model behaviour.

When two particles approach, contact is established when they physically touch, i.e., at point (3) in Figure 4. At this point, the force snaps from zero to $8F_{po}/9$ in magnitude. With further loading (compression), the force decreases in magnitude until the point of equilibrium is reached (point 4 in Figure 4). Further loading results in a compressive force, pushing the particles apart. Unloading follows the same path to point (4) and point (3). Then, as the particles move apart, the contact area decreases, with a *neck* forming. During this stage, the tensile force is still increasing in magnitude, as shown between points (3) and (2) in Figure 4. The maximum tensile force $F_{po}$ occurs at some negative overlap point (2) corresponding to $F_n^{JKR}/F_{po} = 1$ with $a/a_0 = 1/2^{2/3}$. On further unloading, the tensile force rapidly decreases in magnitude until complete detachment occurs at $\delta_n/\delta_{to} = 1$, where the force snaps to zero.

The difference in the loading and unloading paths (non-reversible loading-unloading behaviour) results in the energy dissipation, referred to as the adhesion energy. The energy dissipated can be calculated by the area under the load-displacement curve where the overlap is negative, i.e., between points (3), (2) and (1) in Figure 4.

The normal contact force is calculated using Equation (24), which takes the contact patch radius, *a*, as input. However, this contact radius is not readily available in the DEM formulation, which is based on the normal overlap $\delta_n$. For this contact model, the relation between the contact area *a* and normal overlap $\delta_n$ is given by the non-linear relation in Equation (29). However, solving Equation (29) for *a* is not trivial. Nonetheless, two approaches have emerged.

The method used to create Figure 4 uses a precomputed array for solving Equation (29), which is then used as a look-up table and interpolated at run-time [40]. This method normalises the relations between the force and the patch radius (Equation (24)) using the $F_n^{JKR}/F_{po}$ and $a/a_0$ ratios. In addition, the relation between the overlap and the patch radius is normalised using the $\delta_n/\delta_{to}$ ratio. This resulting set of equations is given by Chokshi et al. [55] and Marshall [40],

$$\frac{F_n^{JKR}}{F_{po}} = 4\left(\left(\frac{a}{a_0}\right)^3 - \left(\frac{a}{a_0}\right)^{\frac{3}{2}}\right) \qquad (32)$$

$$\frac{\delta_n}{\delta_{to}} = 6^{\frac{1}{3}}\left(2\left(\frac{a}{a_0}\right)^2 - \frac{4}{3}\left(\frac{a}{a_0}\right)^{\frac{1}{2}}\right) \qquad (33)$$

Secondly, Parteli et al. [57] derived a closed-form analytical solution to calculate *a* for a given $\delta_n$. The solution expresses Equation (29) as a fourth-order polynomial and subsequently solves its roots (see Deng et al. [58] and Parteli et al. [57]).

### 3.5. Simplified JKR Models (SJKR)

The JKR model from Section 3.4 does not allow for trivial implementation in DEM as the contact patch radius $a$ is not readily available, and the relation between the contact overlap, $\delta_n$, and $a$ is non-linear and time-consuming to solve (Equation (29)). The latter gave rise to several simplifications, which have subsequently become known as the *simplified* JKR models (SJKR). To clearly distinguish between these simplified models and the original JKR model, the term *full* JKR is often used to refer to the original model described in Section 3.4.

The simplification essentially splits Equation (24) into an elastic and an adhesive component. It further assumes that the contact area of the elastic component is *unaffected* by the adhesive force (contrary to the JKR theory). However, one of the earliest simplifications was proposed by Matuttis and Schinner [59], who made use of a generalised formulation for the normal force,

$$F_n^{SJKR} = F_n^H + F_n^a$$

$$\text{with: } F_n^H = k_n \delta_n^{3/2} \text{ – Hertzian force component } [\text{N}]$$

$$k_n = \tfrac{4}{3}\sqrt{R^*}E^* \text{ – Normal elastic stiffness } [\text{N/m}^{3/2}]$$

$$F_n^a = C_0 A_c \text{ – Generalised adhesive (tensile) force component } [\text{N}]$$

$$C_0 \text{ — Cohesion energy density per unit volume } [\text{J/m}^3]$$

$$A_C \text{ — Cohesion contact area } [\text{m}^2]$$

(34)

where the elastic force $F_n^H$ is identical to that of the Hertz-Mindlin contact model often used in DEM, and the cohesion parameter $C_0$ is called the *cohesion energy density* with units of J/m³.

Numerous formulations have been utilised in literature to approximate the larger cohesive contact area $A_c$. However, these are not used consistently, and it can be difficult to discern the exact formulation used when comparing different studies that employed these models. Based on the literature, the naming convention is also not clear and different names are used, often for what appears to be identical implementations, e.g., simplified JKR, SJKR, SJKR1 and SJKR2. Accordingly, to avoid ambiguity, a new naming convention is proposed here using the names SJKR-A to SJKR-F. Furthermore, the comparison of the various SJKR model's force-displacement curves is shown in Figure 5.

The formulation of the simplified models is given in Sections 3.5.1–3.5.6, and the literature where these models were used is reviewed in Section 3.5.7 and categorised under the new naming convention.

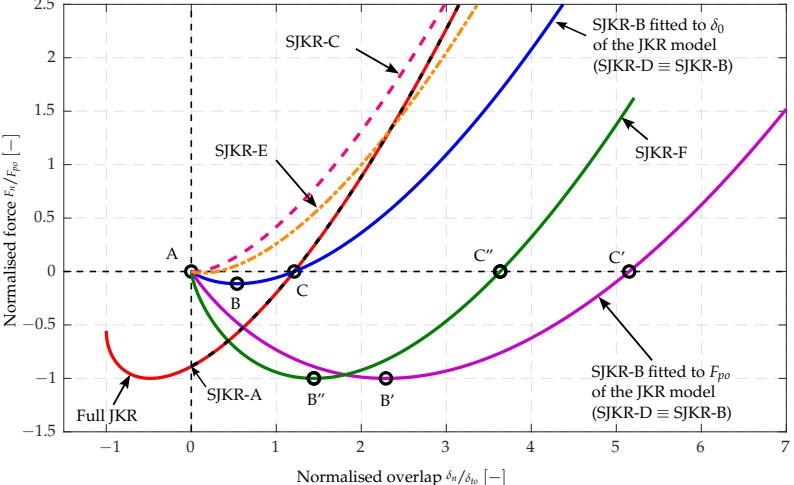

**Figure 5.** The load-displacement curve of the full JKR, SJKR-A, SJKR-B, SJKR-C, SJKR-D, SJKR-E and SJKR-F models. The parameters are normalised using $F_{po}$ and $\delta_{to}$ from the full JKR theory (Section 3.4).

### 3.5.1. The SJKR-A Model

The first simplified JKR model is referred to as the SJKR-A model. This model is identical to the full JKR model (Section 3.4), except that it does not allow negative overlap (a gap between the particles during contact) and sets the contact force to zero when $\delta_n < 0$. In other words, the full JKR formulation is used, but contact is established and broken at zero overlap. This allows for a somewhat simpler implementation as the difference in loading and unloading behaviour does not have to be considered. However, where the full JKR model dissipates energy due to this non-reversible loading-unloading behaviour, the SJKR-A model does not. Also, Equation (29) still needs to be solved, which the other SJKR models completely avoid. The load-displacement curve of the SJKR-A model is shown in Figure 5 alongside that of the full JKR model.

### 3.5.2. The SJKR-B Model

The second simplified JKR model calculates the contact area as,

$$A_c = \pi a^2 = 4\pi R^* \delta_n \tag{35}$$

The relation $a^2 = 4R^* \delta_n$ is assumed, which results in $A_c$ being four times larger than Hertz's contact area. Consequently, Equation (34) can be written in terms of the normal overlap $\delta_n$,

$$F_n^{SJKR} = \frac{4}{3}\sqrt{R^*}E^*\delta_n^{\frac{3}{2}} - 4\pi C_0 R^* \delta_n \tag{36}$$

Examining Figure 5 shows that the SJKR-B model always establishes, and breaks contact at zero overlap $\delta_n = 0$, where the contact force is $F_n^{SJKR} = 0$ (point A in Figure 5). Furthermore, the overlap at the equilibrium point ($F_n^{SJKR} = 0$) may be found as,

$$\delta_0 = R^* \left( \frac{3\pi C_0}{E^*} \right)^2 \tag{37}$$

Subsequently, the energy density may be written as a function of the overlap at the equilibrium point,

$$C_0 = \frac{E^*}{3\pi}\sqrt{\frac{\delta_0}{R^*}} \tag{38}$$

Some form of equivalence may be obtained between the full JKR and the SJKR-B models by selecting the value of $C_0$ such that $\delta_0$ is equal for both models (point C in Figure 5). Importantly, this will result in the magnitude of the maximum tensile (pull-off) force $F_{po}$ at point B being much smaller than that of the full JKR model (Equation (27),

$$F_{po} = \frac{16}{3}(\pi C_0)^3 \left( \frac{R^*}{E^*} \right)^2 \tag{39}$$

Alternatively, the value of $C_0$ can be selected so that the pull-off force (Equation (39) is equal to that of the full JKR model (point B' in Figure 5),

$$C_0 = \frac{1}{\pi}\left( \frac{E^*}{R^*} \right)^{\frac{2}{3}}\left( \frac{3F_{po}}{16} \right)^{\frac{1}{3}} \tag{40}$$

where $F_{po}$ is set to that of the full JKR model (Equation (27)). However, in this case, the $\delta_0$ will be much greater than that of the full JKR (point C' versus point C in Figure 5).

### 3.5.3. The SJKR-C Model

The third model is similar to the SJKR-B model and also based on Equation (34). The only difference is that the calculation of the contact area is,

$$A_c = \pi a^2 = \pi R^* \delta_n \tag{41}$$

which assumes the Hertz relation $a^2 = R^* \delta_n$, not only for the Hertzian force component but also for the adhesive force component. The force (Equation (34)) can then be written as:

$$F_n^{SJKR} = \frac{4}{3}\sqrt{R^*} E^* \delta_n^{\frac{3}{2}} - \pi C_0 R^* \delta_n \tag{42}$$

Consequently, the only difference between the SJKR-B and SJKR-C models is the adhesive force component's contact patch area being four times smaller. As such, all of the relations derived for the SJKR-B model apply to the SJKR-C model if the cohesive energy density is scaled by a factor of four. For example, consider the cohesion energy density, $C_0$, employed when fitting the SJKR-B model to the equilibrium point C in Figure 5—utilising this $C_0$ value results in the SJKR-C model's contact force being greater than that of the SJKR-B model over the whole range of dimensionless ratios examined in Figure 5. Moreover, using the same cohesion energy density, the SJKR-C's pull-off force is smaller than that of the SJKR-B model by a factor $4^3 = 64$.

### 3.5.4. The SJKR-D Model

In this model, the adhesion contact area is given by,

$$A_c = \pi a^2 = 2\pi R_{min} \delta_n \tag{43}$$

As such, the contact area is determined from the minimum radius $R_{min}$ of the two contacting bodies and the contact force is then given by (Equation (34)),

$$F_n^{SJKR} = \frac{4}{3}\sqrt{R^*} E^* \delta_n^{\frac{3}{2}} - 2\pi C_0 R_{min} \delta_n \tag{44}$$

Further comparisons between the SJKR-B, SJKR-C and SJKR-D models may be elucidated by assuming a mono-dispersed system with all particle radii equal to $R_p$. Hence, the effective radius of curvature $(R^*) = R_p/2$. The latter will result in the contact patch radius of the SJKR-B model being $a^2 = 2R_p \delta_n$. Congruently, the contact patch radius of the SJKR-C model and the SJKR-D model will be $a^2 = 1/2 R_p \delta_n$ and $a^2 = 2R_p \delta_n$, respectively. Thus, for mono-dispersed particles, the adhesion force component of the SJKR-D model (assuming the same $C_0$) is equal to that of the SJKR-B model and differs by a factor of four from that of the SJKR-C model.

### 3.5.5. The SJKR-E Model

The calculation of the adhesion contact area for this model is given by,

$$\begin{aligned} A_c = \pi a^2 &= \frac{\pi}{4d^2}(d - R_1 - R_2)(d + R_1 - R_2)(d - R_1 + R_2)(d + R_1 + R_2) \\ &= \frac{\pi}{4d^2}\left(4d^2 R_1^2 - \left(d^2 - R_2^2 + R_1^2\right)^2\right) \end{aligned} \tag{45}$$

where $d$ is the centre distance between two particles with radii $R_1$ and $R_2$, respectively. Accordingly, the contact patch area is determined through simple geometry from the circular lens region that forms when two spheres overlap. Consequently, the contact force becomes (Equation (34)),

$$F_n^{SJKR} = \frac{4}{3}\sqrt{R^*} E^* \delta_n^{\frac{3}{2}} - \frac{C_0 \pi}{4d^2}\left(4d^2 R_1^2 - \left(d^2 - R_2^2 + R_1^2\right)^2\right) \tag{46}$$

Further scrutiny indicates that Equation (45)'s area is approximately equal to $2\pi\delta_n R^*$. That is, for overlaps ranging from 0 to 5% of the $R^*$ and the ratio $R_2/R_1$ spanning from 1 to 10, the difference is less than 2%. As such, the contact area of the SJKR-E model is approximately twice the contact area of the SJKR-C model, half of the SJKR-B model's area and (for mono-disperse particles) also half of the SJKR-D model's contact area. Subsequently, the same SJKR contact force can be achieved amongst the SJKR-B through SJKR-E models by scaling the $C_0$ according to these ratios.

For comparative purposes, Figure 5 shows how the same $C_0$ utilised to fit the SJKR-B model to point C influences SJKR-E's total contact force. The force is smaller in magnitude than that of SJKR-C and larger than that of SJKR-B for the entire range of normalised overlaps. Moreover, SJKR-E's pull-off force is smaller by a factor $2^3 = 8$ (Equation (39)) compared to that of the SJKR-B model.

### 3.5.6. The SJKR-F Model

Recall from Section 3.5 that the simplified JKR models attempt to overcome the difficulties of solving Equation (29) by somehow relating particle overlap to a larger cohesive contact area than the Hertzian contact area. Subsequently, this larger cohesive contact area is employed to calculate the cohesive force component and add it to the Hertz elastic force component.

However, Del Cid [60] assumed the contact area of the JKR model to be equal to that of Hertz's theory and substituted $a^2 = R^*\delta_n$ into the original JKR formulation (Equation (24)) to obtain the contact force expressed in terms of the $\delta_n$ as follows,

$$F_n^{SJKR} = \frac{4}{3}E^*\sqrt{R^*}\delta_n^{\frac{3}{2}} - \sqrt{8\pi\gamma^*E^*}(R^*\delta_n)^{\frac{3}{4}} \tag{47}$$

This approach eliminates the need to first solve Equation (29) to get the contact radius from a given overlap. The pull-off (maximum tensile) force is given by $F_{po} = 3/2\pi\gamma^* R^*$ (point B″ in Figure 5), which is the same as that of the full JKR model (Equation (27)) and occurs at an overlap,

$$\delta_{po} = \left(\frac{9\pi\gamma^*}{8E^*}\right)^{\frac{2}{3}} R^{*\frac{1}{3}} \tag{48}$$

The equilibrium (zero) force is, however, at a larger overlap (point C″ in Figure 5) when compared to that of the full JKR theory (assuming the same surface energy) and is given by,

$$\delta_0 = \left(\frac{9\pi\gamma^*}{2E^*}\right)^{2/3} R^{*\frac{1}{3}} \tag{49}$$

This model uses the somewhat more tangible surface energy density ($\gamma^*$) cohesive parameter, whose units are J/m² and not the less palpable cohesion energy density ($C_0$) parameter, with the units J/m³. Also, when using the same value for $\gamma^*$, the SJKR-F model has the same pull-off force as that of the full JKR model and an equilibrium overlap (point C″ in Figure 5), which is not as large as that of the SJKR-B model (point C′) with $C_0$ selected to have the same pull-off force.

### 3.5.7. SJKR Implementations

Coetzee [61] implemented the SJKR-A model as a user-selected option and part of the full JKR model for use in PFC [36], whilst implementation of the subsequent SJKR models by Coetzee is documented in [62].

The SJKR-B model is implemented in LIGGGHTS [63] and called the *SJKR2* model. The SJKR-C model is implemented in EDEM [50] and called the *Linear Cohesion V2* model. The SJKR-D model was implemented in EDEM [50], as an earlier version of the *Linear Cohesion V2* (SJKR-C) model, and called the *Linear Cohesion* model. This model, however, is not appropriate for non-uniform particle size distributions since the minimum particle

radius will always be used in the force calculation [50]. However, this implementation in EDEM was used by Grima [64] and Grima and Wypych [65] to model the cohesive behaviour of wet coal and bauxite. The SJKR-E model is implemented in LIGGGHTS [63], where it is referred to only as the *SJKR* model.

The SJKR-F model was implemented by Del Cid [60] and Umer and Siraj [66] and called the *simplified JKR* model. Hashibon et al. [67] used Equation (47) (SJKR-F) to model history-dependent powder flow and scaled the contact stiffness and the surface energy based on the load history. Although sliding and rolling friction were not included, Hashibon et al. [67] concluded that this model, at least qualitatively, accurately predicted the flow behaviour of powder discharging from a rectangular bin. Umer and Siraj's implementation modelled the capillary forces in a wet poly-dispersed granular flow of glass beads moving over a single mixing blade.

Elmsahli [68] and Elmsahli and Sinka [69] stated that they implemented the 'original' JKR model in LIGGGHTS. However, upon further scrutiny and studying their load-displacement curves, and despite referring to Equation (24) (full JKR model), it is clear that they implemented the SJKR-F model. Note that although Elmsahli [68] and Elmsahli and Sinka [69] do not refer to the work by Del Cid [60], they use the same basic formulation given by,

$$F_n^{SJKR} = \sqrt{R^*}E^*\delta_n^{3/2} - \sqrt{6\pi\gamma^*E^*}(R^*\delta_n)^{3/4} \tag{50}$$

where the constant coefficients differ from that derived in Equation (47). They used this model to investigate the effects of particle shape and size, sliding and rolling friction and cohesion (surface energy) on the packing ratio of powders. This study aimed to develop a model capable of predicting low packing densities (high porosities in the order of 0.7) often observed in typical powders.

### 3.6. Liquid-Bridge Contact Model

The fundamental mechanisms responsible for cohesion that is observed for wet materials are described by Newitt and Conway-Jones [25]. Essentially, the cohesive force arising from the formation of a liquid bridge is the sum of two forces. The first is the surface tension ($\gamma$) arising from the air-liquid interface, and the second is the resultant force due to hydrostatic pressure differences between the liquid and the air. An illustration of the simple particle-particle liquid-bridge force corresponding to Newitt and Conway-Jones's description is presented in Figure 6, with the gap between the contacting pieces indicated by $\delta_d$. As such, the negative normal overlap would occur, and a negative (attractive) force would act on the particles. However, non-DEM-related literature typically expresses the liquid-bridge force as a positive value. As such, for uniformity with the literature's formulations, the separation distance is defined as $\delta_d = -\delta_n$ when $\delta_n \leq 0$.

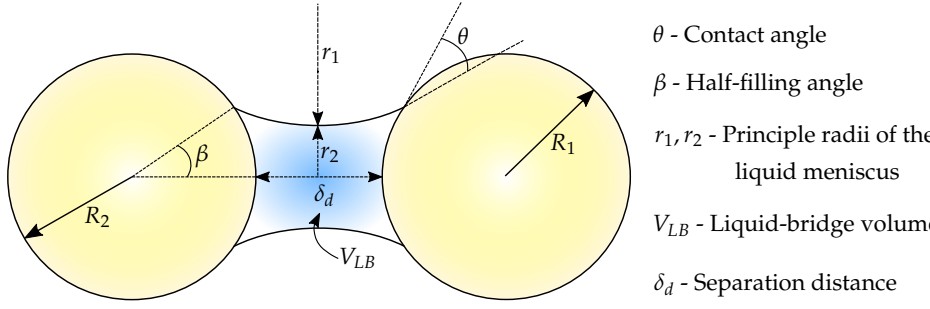

$\theta$ - Contact angle

$\beta$ - Half-filling angle

$r_1, r_2$ - Principle radii of the liquid meniscus

$V_{LB}$ - Liquid-bridge volume

$\delta_d$ - Separation distance

**Figure 6.** A schematic representation of the liquid-bridge contact model.

Unfortunately, the Young-Laplace equation describing the pressure difference and the shape of the air-fluid interface does not have a closed-form solution. As such, numerical methods will have to be employed to determine the geometry of the liquid bridge, which is impractical for DEM modelling. However, several researchers have developed approximate formulations through assumptions and linear regression of the numerical solution or

experimental measurements. A review of various model implementations for the pendular regime [70–73] can be found in Gladkyy and Schwarze [74]. Importantly, Gladkyy and Schwarze [74] conclude that although liquid content has a marked effect on the granular material, the specific choice of model is of 'minor importance'. Critically, the various liquid-bridge models comprise two fundamental aspects: the rupture distance $D_0$ and the rupture (pull-off) force $F_0$. The former depends on the volume of the liquid-bridge $V_{LB}$, while the latter is greatest when the separation distance $\delta_d$ is zero.

Figure 7 shows the force-displacement relation in the normal direction. When two pieces approach each other, the path along 1 is followed, and contact is only established once the two pieces make physical contact at point 2a ($\delta_n = \delta_d = 0$ and ignoring the thickness of the liquid film). The contact force then jumps to the value of the pull-off force at point 2b. Further loading follows the path from 2b to 3, and unloading is along the 3-2b-4 path until the rupture distance is reached at point 5a. At this point, the bridge ruptures, and the force jumps to a value of zero at point 5b.

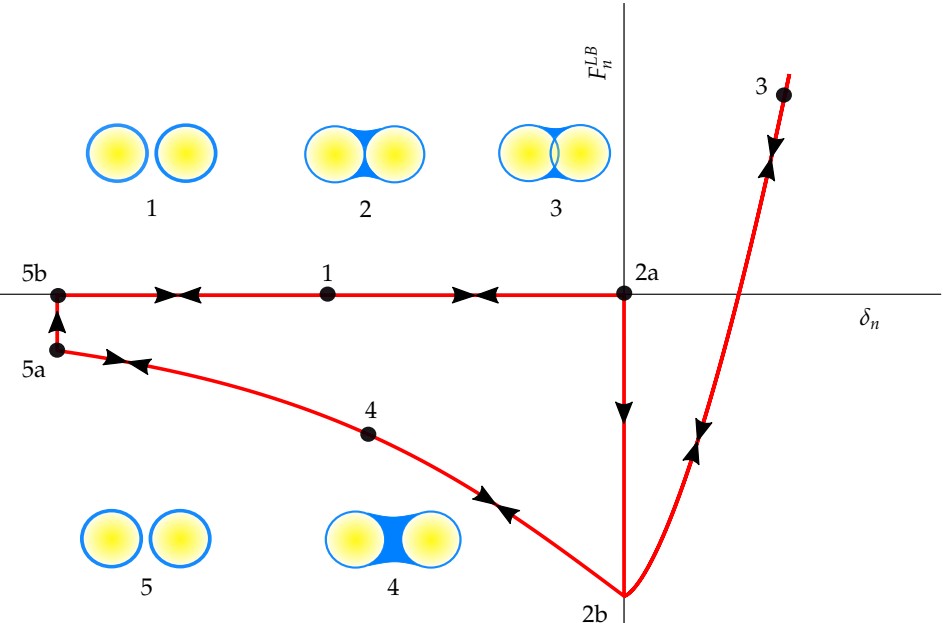

**Figure 7.** General normal force-displacement law of the liquid-bridge model.

### 3.6.1. Rupture Distance

The first fundamental component of the liquid-bridge model comprises the rupture or critical separation distance. The force becomes zero when $\delta_d$ between two contacting pieces exceeds this critical distance. Fundamentally, this distance is dependent on the cube root of the bridge volume $V_{LB}$. The shape of this liquid volume is illustrated in Figure 6 between two spheres of equal size, and the rupture distance is given by Lian et al. [75],

$$D_0 \simeq \left(1 + \frac{\theta}{2}\right) \sqrt[3]{V_{LB}} \tag{51}$$

Other formulations for the rupture distance are listed by Coetzee [76], which take spheres of unequal radii and separation velocity into account. However, all are fine distinctions with core dependence on $\sqrt[3]{V_{LB}}$. Conventionally, researchers consider the contact angle very small [77] with $\theta \to 0°$. This negligible influence of the contact angle results in the rupture distance simplifying to $D_0 \approx \sqrt[3]{V_{LB}}$, which agrees with the conclusion by Lian et al. [75] for contact angles of $< 40°$.

### 3.6.2. Cohesive Capillary Force

The second fundamental component of the liquid-bridge model is the maximum attractive-tensile (rupture) force $F_0$, which occurs at zero separation distance [71–73],

$$F_0 = 2\pi R^* \gamma^* \cos(\theta) \tag{52}$$

The force-displacement relation for particle-particle contact is given by Lambert's formulation [73],

$$F_n^{LB} = -\frac{2\pi R^* \gamma^* \cos(\theta)}{1 + \frac{\delta_d}{2d_{ip}}} \tag{53}$$

with: $d_{ip} = \frac{\delta_d}{2}\left(-1 + \sqrt{1 + \frac{2V_{LB}}{\pi R^* \delta_d^2}}\right)$ – Particle-particle immersion depth [m]

and that of particle-wall contact follows as,

$$F_n^{LB} = -\frac{2\pi R^* \gamma^* \cos(\theta)}{1 + \frac{\delta_d}{d_{iw}}} \tag{54}$$

with: $d_{iw} = -\delta_d + \sqrt{\delta_d^2 + \frac{V_{LB}}{\pi R^*}}$ – Particle-wall immersion depth [m]

Various nuance distinctions exist between the rupture path formulations of the numerous liquid-bridge models (see [70–73,78–81]). For instance, Lambert's formulation [73] is a simplification of Rabinovich's formulation [72]'s. However, all formulations fundamentally depend on the product of a particle length and surface tension and follow a convex hyperbolic unloading path. Accordingly, these distinctions are of minor importance when the simplifications incorporated in bulk-level DEM modelling are considered. Further details regarding the intricacies of the liquid-bridge models are available in Coetzee [76].

### 3.7. Gilabert's Linear Cohesive Model

Gilabert et al. [82] proposed a simple and generic linear cohesive model and described it as: '*A linear approximation of a realistic van der Waals force law and contains two essential parameters: maximum attractive force $F_0$ and range $D_0$*'. Furthermore, this model can also be viewed as a linearization of the liquid-bridge model (Section 3.6), where $F_0$ can be taken as the product of a surface tension $\gamma^*$ and a particle length $R^*$ when liquid menisci are modelled.

The normal force is given by,

$$F_n^{LC} = \begin{cases} k_n \delta_n + \eta_n \dot{\delta}_n + F_0 & \text{if} & \delta_n \geq 0 \\ F_0(1 + \delta_n/D_0) & \text{if} & D_0 < \delta_n < 0 \\ 0 & \text{if} & \delta_n \leq D_0 \end{cases} \tag{55}$$

with: $F_0$ — Maximum attractive force [N]

$D_0$ — Rupture threshold [m]

An illustration of the model behaviour is depicted in Figure 8. The shear component follows the linear contact model's customary Coulomb-type friction law. However, the adhesive component $F_0$ influences the Mohr-Coulomb failure envelope. This is demonstrated in Figure 8b, with $F_0$ shifting the tip of the Mohr-Coulomb cone away from zero.

The maximum tensile force occurs at zero overlap, as observable in Figure 8a, which is equivalent to the liquid-bridge force (Figure 7). This property makes the model suitable for the modelling of liquid bridges under the assumption of linear behaviour. In summary, the model only requires the specification of a maximum tensile force and rupture distance alongside the parameters of the well-known linear (non-cohesive) contact model [33].

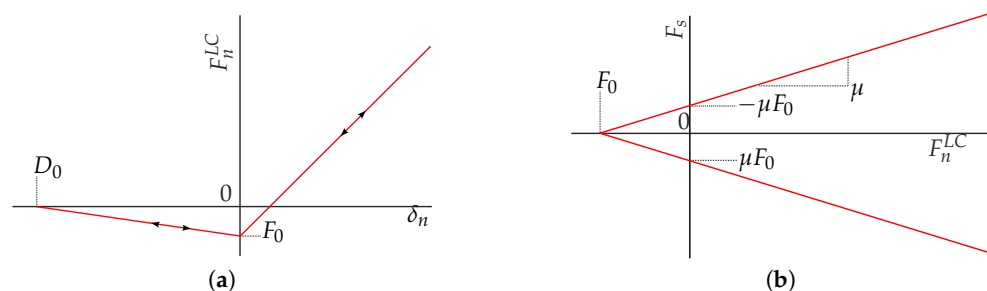

**Figure 8.** Force-deformation behaviour of the linear cohesive contact model (adapted from [82]). Reproduced with permission from Physical Review E; published by American Physical Society, 2007. (**a**) Normal force (damping omitted). (**b**) Coulomb limit of the shear force.

### 3.8. Tangential and Damping Contact Models for Cohesive Materials

As discussed in Section 3.1, the Coulomb friction limit is usually modified to account for cohesive effects in the tangential (shear) direction. The details of each tangential model are not presented, but in general, the following procedure is used where the force vector is incrementally updated,

$$\boldsymbol{F_s^*} = \boldsymbol{F_s^t} + k_s \boldsymbol{\Delta \delta_s} \tag{56}$$

where $\boldsymbol{F_s^t}$ is the tangential force at the end of the previous time-step $t$, $\boldsymbol{\Delta \delta_s}$ is the relative tangential displacement increment, and $k_s$ is the shear stiffness which can be constant [43] or a function of the normal overlap [49]. The following criteria limits the updated force from Equation (56),

$$\boldsymbol{F_s^{t+1}} = \begin{cases} \boldsymbol{F_s^*} & \text{if } |\boldsymbol{F_s^*}| \leq \mu(F_n - F_{po}) \\ \mu(F_n - F_{po})\left(\frac{\boldsymbol{F_s^*}}{|\boldsymbol{F_s^*}|}\right) & \text{otherwise} \end{cases} \tag{57}$$

where $F_n$ is the normal force and the minimum force, $F_{po}$, is the maximum tensile force. Note that $F_{po}$ has a negative value and will thus always increase the reference force $(F_n - F_{po})$, which can never be smaller than zero.

For further implementation details of the tangential models and that of the viscous damping models, the reader is referred to the documentation by Coetzee [48,61,62,76,83] and the references therein.

## 4. Parameter Calibration of Cohesive Materials

This section critically reviews studies on the calibration of DEM parameters for cohesive bulk materials. The focus is on applications in the mining and agricultural sectors where cohesion is primarily due to moisture (water content) and not necessarily the pharmaceutical and food industries where powders or small particles and van der Waals effects are present.

The work is categorised into ten methodologies according to the use of a specific experiment, series of experiments, or systematic calibration procedures. A given method is not necessarily by a particular author, group of authors or research group but rather a collection of work with a similar theme or approach. This review provides an overview of experiments from which researchers and DEM practitioners can select. For the detailed design, geometric dimensions and execution of the experiments, readers should consult the relevant literature. That said, it is important for the DEM model to closely resemble the experimental setup in geometry, dimensions and techniques (such as filling, for example), even though the specific geometry and dimensions may vary from case to case. Furthermore, all the studies presented here conducted three-dimensional simulations, although it might be equally applicable to two-dimensional modelling if the experiment's flow is predominantly two-dimensional.

Methodology 1 to Methodology 5 looked at a specific experiment to calibrate one or more parameters, while Methodology 6 and 7 combined two or more experiments. Method-

ology 8 looked at calibrating a highly polydisperse and cohesive material. Methodologies 1 to 8 looked at calibrating the parameter values for a cohesive material directly, in other words, simultaneously calibrating all parameters (cohesive and non-cohesive). On the other hand, Methodology 9 and Methodology 10 first calibrated the parameter values for the material in a non-cohesive (dry) state and then introduced and calibrated the cohesive parameters while keeping the non-cohesive parameters primarily unchanged. Therefore, these two methodologies are also considered the most systematic in their approach and procedures. In addition, Methodology 10 introduced a new technique for calibrating materials with a relatively high level of cohesion, which would, under normal conditions, not flow in a test where an orifice or opening of some kind is used or where an angle (shear or repose) other than vertical, which is sensitive to the level of cohesion, is a requirement.

### 4.1. Methodology 1: Draw Down Tests

The draw-down test is a relatively simple test that can measure a number of bulk materials and flow properties. An example of such a setup is shown in Figure 9, with the upper box and lower box indicated. The upper box is filled with granular material, and the trapdoor separating the two boxes is opened to allow the material to flow freely into the lower box. The remaining material in the upper box forms two angles, referred to as the shear angle (taking the average of the two angles). The material in the lower box forms a pile defined by another two angles, called the angle of repose (AOR, taking the average of the two angles). Additionally, the instantaneous mass flow rate can be measured if the setup is equipped with load cells, or the total discharge time (average flow rate) can be measured. Although related to the angles, the mass that remains in the upper box or the discharged mass as a ratio of the total mass can also be measured. As such, the draw-down test provides two to four bulk measures, which can be used for calibration purposes and was successfully used for non-cohesive materials (see [7,13,15,84,85]).

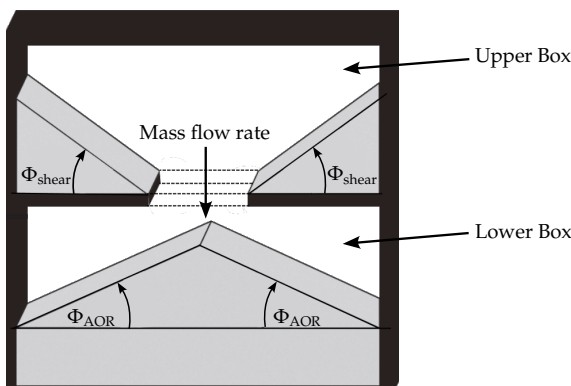

**Figure 9.** A typical draw-down test setup.

Although ideal for non-cohesive and slightly cohesive materials, the standard draw-down test and procedures are not ideal for cohesive materials that arch (block) in the opening and produce steep shear angles close to vertical. However, Ajmal et al. [86] showed that it could be used to calibrate a DEM model for cohesive sand with a 10% moisture content if the methodology used for non-cohesive materials is slightly adapted. Using the full JKR and an SJKR (specific model not specified) contact model, the sliding and rolling friction coefficients and the cohesion parameter were calibrated. In addition, an elasto-plastic-dashpot rolling resistance model was implemented with the SJKR model, while an adhesive hysteretic rolling resistance model was implemented with the full JKR model. The calibration was limited to materials under 'rapid flow and low consolidation' conditions, and the physical particle size was scaled up by a factor of 10 in the model.

It was shown that the blockage or arching (no flow) of material at the opening was independent of the coefficients of sliding and rolling friction and only dependent on the cohesion parameter. Comparing reproducible no-flow and sure-flow results to experimental

observations for various opening sizes, a narrow range of possible values for the cohesion parameter could be established. A unique set of parameter values could be obtained by comparing the angle of repose formed in the lower box and the total mass that flowed into the bottom box to experimental measurements. It was noted that the shear angle formed in the upper box could not be used since it was not reproducible and dependent on the amount of material that flowed out of the upper box. The mass flow rate was also inconsistent (the material would slump down), with the total mass transfer used instead.

The two contact models were calibrated and compared to the experimental measurements and observations. It was found that the SJKR model (with elasto-plastic-dashpot rolling resistance) accurately predicted the shear angle and the discharged mass but over-predicted the angle of repose by approximately 30%. On the other hand, the full JKR model (with adhesive hysteretic rolling resistance) was more accurate in predicting the angle of repose with an error of 15% while showing similar results for the shear angle and discharged mass.

It was further shown that it is important to accurately calibrate not only the particle-particle cohesion but also the particle-wall adhesion. Using the same cohesion energy density for particle-wall and particle-particle contacts in the SJKR model produced satisfactory results. However, doing the same with the full JKR model resulted in the particle-wall adhesive behaviour being significantly over-estimated (too many particles got stuck to the walls). As a result, the surface energy of particle-wall contact had to be set equal to half the value of particle-particle contact. The authors attributed some of the different behaviour to using the adhesion hysteretic rolling resistance model. However, this model was only implemented with the full JKR model and not with the SJKR model. Similarly, the elasto-plastic-dashpot rolling resistance model was only implemented with the SJKR model and not with the full JKR model, which makes direct comparisons difficult.

Ajmal et al. [86] did not investigate this further, but it is our judgement that the difference in model behaviour is not only due to the different rolling resistance models but also due to the fundamental differences between the full JKR and SJKR models. For example, contrary to the SJKR models, the full JKR model is non-reversible. That is, the load-displacement curve does not follow the same path on loading and unloading, which results in a loss of energy (energy of adhesion, see Section 3.4). Also, in the SJKR models, the contact force returns to zero upon complete unloading, while in the full JKR model, a tensile force remains until the tear-off distance (gap) is reached and the force snaps to zero.

For the calibration of cohesive iron ore using a draw-down test and a JKR contact model see Carvalho et al. [87] and using a combination of the liquid-bridge and SJKR-E contact models, see Carr [88] and Nase et al. [32].

The draw-down test has the advantage of being simple and easy to design, manufacture, execute, and model. Even if image processing software is unavailable, the shear angle and the angle of repose can be manually measured. More sophisticated equipment, such as load cells, is needed to accurately measure the mass flow rate, but if this is not available, the average mass flow rate can be calculated using a stopwatch. Although not ideal for precision measurements, this will provide some information which can be used in the calibration process. The setup can easily be scaled to accommodate larger particles, but one disadvantage is that the side panels should be transparent, which can be problematic and expensive if a relatively large setup is used. Another disadvantage of the draw-down test is that the material might block the discharge opening, and even though this phenomenon can be used to identify a parameter range (as done by Ajmal et al. [86] ), it might be problematic if the material is highly cohesive. The use of a draw-down test in combination with a centrifuge (see Section 4.10) should, in future, be investigated.

### 4.2. Methodology 2: Lifting Cylinder Tests

The well-known lifting cylinder test is often used to measure the AOR of non-cohesive and slightly cohesive materials [12,89,90]. An open-ended cylinder, placed on the floor,

is filled with material and then slowly raised vertically, allowing the material to form a static pile. The angle this pile makes with the horizontal is defined as the AOR. Similar to the draw-down test, this test in its standard form is not ideal for cohesive materials that produce inconsistent angles or angles close to vertical, which are not sensitive to changes in the bulk cohesion (e.g., adding more moisture).

Roessler and Katterfeld [91] used wet sand (9.5% moisture content) and showed that the AOR had very high variability, ranging from 40.4° to 84.3° in repeated experimental tests. However, the parameter values could be successfully calibrated by proposing a new methodology for cohesive materials and using the SJKR-E contact model (Section 3.5.5).

Instead of using the AOR, the macroscopic flow behaviour of the material as it exited the cylinder was analysed. The bulk material column that formed changed in diameter as the cylinder was lifted until the column finally collapsed under its own weight. The column diameter was measured at four different heights, and four distinct and reproducible phases could be identified during the upward motion of the cylinder, as shown in Figure 10. These phases included: the build-up of a stable column, the convex bending of the column, the beginning of the column collapse, and the final resting pile of sand. The convex bending of the bulk column and the moment of the first collapse was proposed as the two calibration criteria and were independent of the lifting velocity and the diameter of the cylinder.

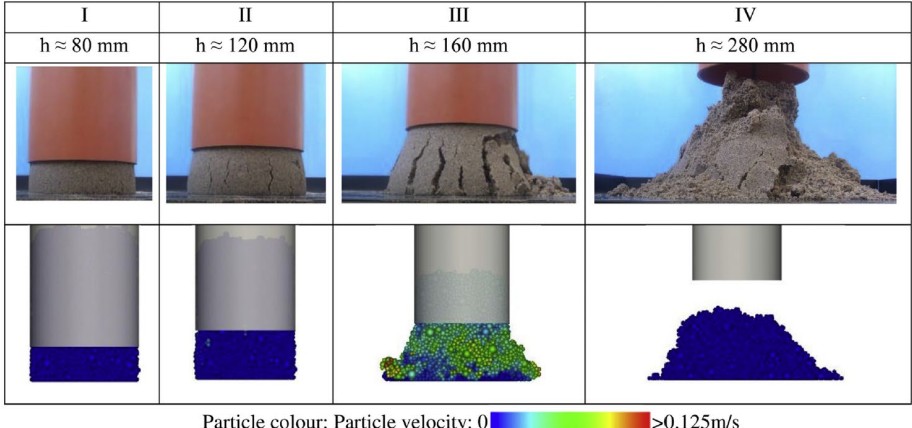

Particle colour: Particle velocity: 0 ▬▬▬▬ >0.125m/s

**Figure 10.** The lifting cylinder test for cohesive material as performed by [91]. Reproduced with permission from Particuology; published by Elsevier, 2019.

Using spherical (upscaled) particles, the coefficients of rolling and sliding friction and the cohesion energy density were the significant parameters that required calibration. The value of each parameter was systematically varied until a calibrated set could be identified. However, this study used a single moisture content, and this approach still needs to be verified for different moisture contents and different bulk materials. Upscaled sand particles were used in the model, and the particle stiffness (shear modulus) was assumed. The combined effects of particle scale, stiffness, and cohesion energy density on the material bulk behaviour need further investigation.

For a recent study where the results from Roessler and Katterfeld [91] were used to identify the interaction between rolling resistance and the cohesive parameter in modelling the bulk behaviour, see Doan et al. [92]. Li et al. [93] also studied the combined effect of rolling resistance and the cohesion parameter of a JKR model on the bulk cohesive behaviour. Moreover, see Yu et al. [94], where the lifting cylinder test was used to calibrate the cohesive parameters of the JKR contact model for the modelling of pig manure. Another study where the lifting cylinder test was used to calibrate cohesive materials was by Nasr-Eddine et al. [95].

The lifting cylinder test is simple and easy to design, manufacture, execute and model. It is scale-invariant as long as the lifting velocity is slow [12,96]. However, it is limited to

non-cohesive and slightly cohesive materials. Highly cohesive materials will not form a pile and will result in steep angles of repose that are not necessarily sensitive to changes in moisture (see Section 4.10 for using a centrifuge to overcome this problem).

Although the lifting cylinder test is the most commonly used angle of repose test, other methods, such as the *pouring* method, are available. Material is poured from a funnel, hopper, or bin at a fixed or variable height onto a pile, from where the angle can be measured; see [93,97–104] for examples. However, compared to the lifting cylinder test, these tests might be even less suited for cohesive materials. For example, the material might block the opening of the container from which the material flows, or the flow might be irregular and inconsistent, which can affect the shape of the pile.

### 4.3. Methodology 3: Free Flow Tests

Chen et al. [105] devised a test where material flows under the action of gravity over several ledges (boxes) and plates, as shown in Figure 11. The test mimics industrial applications such as conveyor transfer chutes. First, the material accumulates in the first two boxes, forming shear angles $\alpha$ and $\beta$, respectively. From here, it flows over a plate at 60° (box 3) and over a plate at 75° (box 4) with the mass $m_1$ and $m_2$ adhering to the plates, respectively. Finally, the material collects in the bottom tray, where two angles of repose $\theta_1$ and $\theta_2$ can be defined. Note that these two angles will not necessarily be equal due to the asymmetric flow.

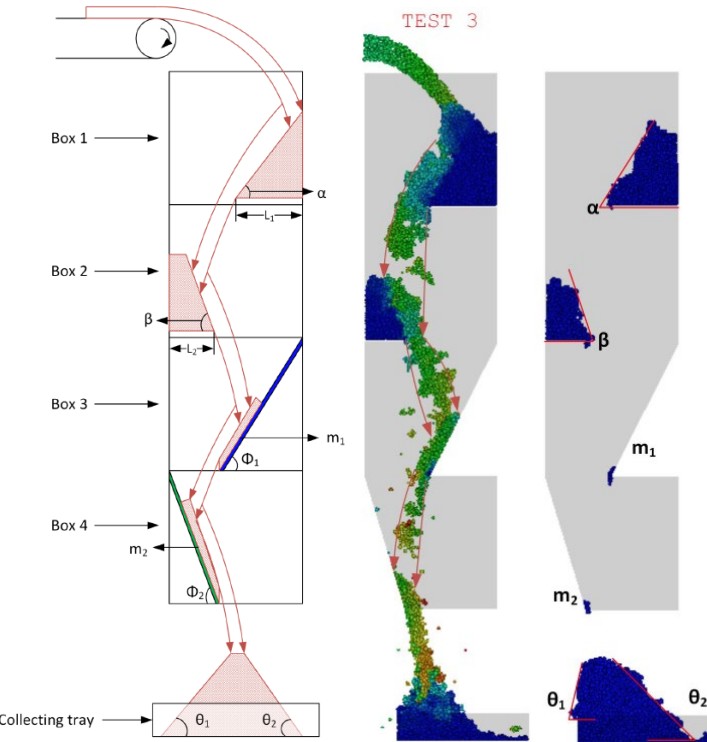

**Figure 11.** Flow test apparatus as adapted from [105]. Reproduced with permission from W. Chen.

The four bulk measures, $\alpha$, $\beta$, $\theta_1$ and $\theta_2$, are reflective of particle-particle contact properties, while the two measures, $m_1$ and $m_2$, are reflective of particle-wall contact properties. This test setup was modelled using the SJKR-E contact model, and the effects of changing the parameter values on the six bulk measures were indicated. Although intended for the calibration of wet iron ore, the calibration of a physical material was not presented as part of the study. e Silva et al. [106] used a similar setup, amongst other tests, to calibrate green iron ore pellets. However, the setup had a funnel at the top, from which the material discharged onto a single inclined plate and then onto the collecting tray.

Carr [88] performed a systematic calibration process for cohesive iron ore (18.5% moisture content), using a ledge and draw-down test in combination with a dynamic adhesion test. A conveyor belt continuously unloaded material onto an inclined impact plate in the latter. For a given angle of the plate, the residual mass and the build-up height perpendicular to the plate were used for calibration. In theory, this test is similar to the first stage of the flow test shown in Figure 11, where box 1 is simply replaced by an inclined plate, and a second conveyor belt recirculates the material that flows from it. Although the residual mass could be accurately modelled, qualitatively, the shape of the build-up material differed from that observed during experimentation. Carr found that for this application, a combination of the liquid-bridge model (Section 3.6) and the SJKR-E model for the particle-particle and particle-wall contacts, respectively, provided the best results. Carr called this the *hybrid* model and also investigated using the SJKR-E model for all contacts. However, in this case, the particle-particle behaviour would be too rigid and more representative of cohesive powder flow than cohesive iron ore.

Carr also investigated the use of the liquid-bridge model for all contacts. However, in the specific implementation, different parameter values for particle-particle and particle-wall contacts could not be specified for this contact model. In addition, the EEPA model was also investigated, and the results seemed promising with particle-particle and particle-wall contacts and bulk behaviour modelled realistically. However, due to computational performance, the EEPA model was not further investigated (see Section 5).

If a hopper or funnel is used to feed the material into the setup shown in Figure 11, it is simple and easy to design, manufacture, execute and model with no moving parts (as used by Quist and Evertsson [107] for a non-cohesive material). In addition, this method has the advantage of measuring particle-particle and particle-wall properties, which can be used to calibrate the various parameters. However, in this case, the material might block the opening of the hopper or funnel if the cohesion is too high. Alternatively, a conveyor belt can feed the material into the flow test apparatus (see Figure 11), but this requires more expensive equipment and complicates the DEM model.

*4.4. Methodology 4: Uniaxial Compression Tests*

Confined uniaxial compression is often used to measure the bulk stiffness of the non-cohesive material [6,8,108] and cohesive material [109]. However, a combination of confined and unconfined compression can be used to obtain the flow properties of cohesive materials. As depicted in Figure 12, a sample of the material is first consolidated under confined conditions in a cylindrical container, and then the container is removed, leaving a free-standing sample, which is then compressed by a platen and the unconfined yield strength measured. The flow function can be obtained using the consolidation pressure and the unconfined strength, which is a much faster and simpler process than using a direct (Jenike) shear tester.

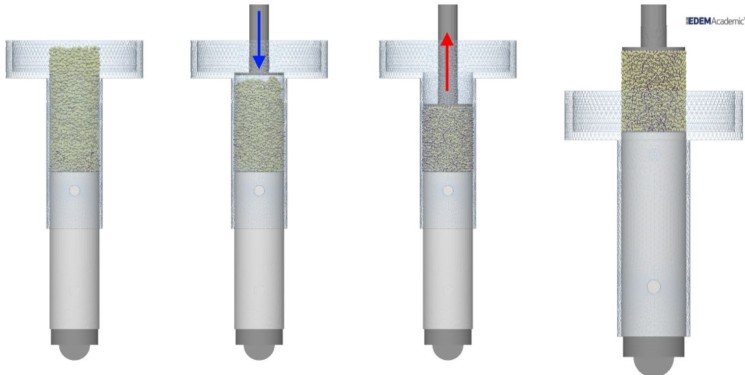

**Figure 12.** Uniaxial compression showing from left to right: filling, confined loading (consolidation), unloading, and unconfined loading as performed by [49]. Reproduced with permission from J. P. Morrissey.

Morrissey [49] developed the EEPA contact model and used a uniaxial compression test to calibrate the EEPA particle-particle contact parameters for moist iron ore fines. The particle-wall interaction was assumed to be non-cohesive and modelled with the Hertz-Mindlin model. The sensitivity of the model to changes in the normal contact stiffness of the loading and unloading paths (Section 3.3), the tangential stiffness, the pull-off force, adhesion energy, the exponent of the adhesion branch $\chi$, the coefficients of sliding and rolling friction, and damping was performed in a detailed and systematic study.

The calibration aimed to obtain a single set of parameter values that would accurately predict the material behaviour for a given level of cohesion (moisture content) over a range of consolidation pressures. For this purpose, experimental data from two flow functions were used. First, two data points were used from the flow function for 1% moisture content, one at 40 kPa and one at 100 kPa consolidation pressure. Next, three more experimental data points were used, each at a consolidation pressure of 100 kPa from the flow functions for 2%, 4% and 6%, respectively, to account for the increased strength with an increase in moisture content.

The modelled particle size and shape (multi-spheres) were selected, and the coefficient of sliding friction was taken from literature as $\mu = 0.5$. A small amount of rolling resistance with $\mu_r = 0.005$ was included to account for the simplified shape representation. Furthermore, the coefficient of restitution was set to 0.5, and the coefficient of sliding friction between the particles and the wall was taken from the literature. The confined compression load-displacement curve was used to calibrate the loading, $k_1$, and unloading, $k_2$, stiffness. Using a single consolidation pressure of 100 kPa for the various moisture contents, the surface energy (adhesion parameter) was calibrated in a regression study.

For more details on using the EEPA contact model, uniaxial compression and scaling of the compression test, see [51,93,110–114].

### 4.5. Methodology 5: Calibration of Highly Consolidated Material

Chen et al. [105] devised a unique test for calibrating materials (iron ore in this case) under high consolidation pressures, see Figure 13. First, the hopper opening width is set to a value $B_1$, resulting in a hopper angle $\alpha$, which is selected such that the material *will* arch. Next, the hopper gate is closed and filled with material, after which a consolidation pressure is applied, as shown. Hereafter, the discharge gate is opened, and the material is allowed to arch. The hopper opening width is then slowly adjusted (relaxing the hopper angle) until all the material has discharged. The final opening width $B_2$ is recorded, the two angles of repose are measured in the collecting tray at the bottom, and the mass of material that adheres to the sides is measured. Subsequently, the whole test is repeated using a different consolidation pressure.

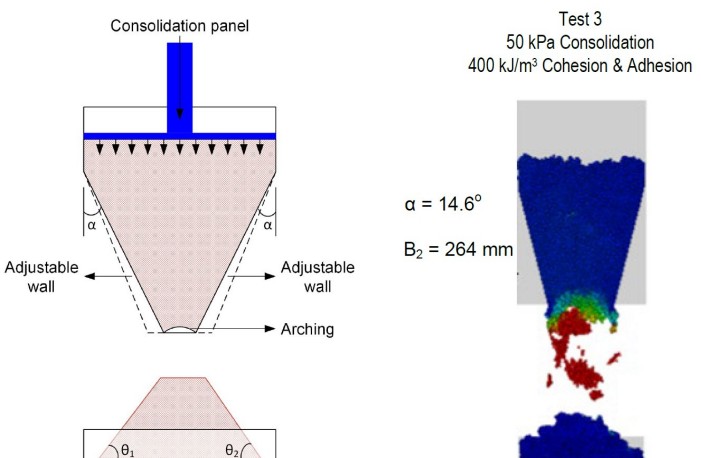

**Figure 13.** Calibration setup for materials under high consolidation pressure as adapted from [105]. Reproduced with permission from W. Chen.

It was demonstrated that the DEM model is sensitive to parameter changes, but experimental results were not presented. Also, the experiment and model might be sensitive to the material-wall interface properties, which should be accurately measured and accounted for in the model.

### 4.6. Methodology 6: Combined Draw Down and Lifting Cylinder Tests

Calibrating multiple unknown parameter values using a single bulk measure, such as the AOR, does not guarantee a unique set of parameter values [13]. Therefore, either a single experiment with more than one measurable material property, such as the draw-down test, or multiple experiments, each with at least one measurable material property, should ideally be used to find a unique set of parameter values.

Katterfeld et al. [9] combined the draw-down test (from Methodology 1) and the lifting cylinder test (from Methodology 2) to calibrate the parameter values for wet gypsum with a 22% moisture content. The EEPA and an SJKR contact model were investigated using the LIGGGHTS [63] software package. However, it is not clear which specific SJKR model was used. Due to the small scale of the gypsum particles (<0.2 mm), upscaled particles were modelled using a size distribution ranging from 7 mm to 10 mm (scale factor > 50). The Young's modulus was chosen as 5 MPa, and the particle-wall coefficient of friction was experimentally measured and directly used in the model. The coefficient of restitution was determined to be 0.2 for the SJKR model and 0.8 for the EEPA model. However, it is not clear how the latter values were established.

For the EEPA model, the exponent of the loading curve, $m$, was determined using a uniaxial compression test. Hereafter, the predicted load-displacement curve was compared to the measured curve, and the exponent was adjusted to obtain the most accurate results. The maximum consolidation load was 6 kPa, and the EEPA model showed a clear advantage in predicting the non-linear load-displacement behaviour compared to the SJKR model.

The combined results of a lifting cylinder test and a draw-down test were then used to calibrate the coefficients of sliding and rolling friction (spherical particles) and the cohesion energy density. In the draw-down test, two opening sizes were used, 150 mm and 160 mm. In the experiments, it was found that the material repeatedly arched (blocked) with the smaller of the two openings while it consistently flowed with the larger opening. This criterion was used to calibrate the parameter values.

Once calibrated based on the criteria described above, both contact models under-predicted the shear angle in the upper chamber and over-predicted the angle of repose in the lower chamber. As a result, the arching behaviour in the upper chamber required a high value for the cohesion energy density, while the angle of repose in the lower chamber required a smaller value.

In an attempt to calibrate the particle-wall adhesion, an experiment where a belt conveyor discharged material onto a Perspex impact plate was used. The adhesion parameter (energy density) was calibrated by qualitatively comparing the general flow behaviour. However, the amount (mass) of material that remained on the impact plate was significantly under-predicted.

The study concluded that, after calibration, several aspects of the behaviour of the wet gypsum could be predicted with sufficient accuracy. However, some differences in the material behaviour still existed, and the model failed to accurately predict the build-up of material in an industrial transfer chute. Katterfeld et al. [9] postulated that this could be attributed to using upscaled particles in the model, which were not accurately capturing the compaction or consolidation of the material.

### 4.7. Methodology 7: Ring Shear, Ledge, Consolidation and Penetration Tests

Direct shear testers, both translational and rotational (ring), have been successfully used to calibrate DEM models for non-cohesive materials (see [6,8]) and slightly cohesive materials, see [93]. However, Mohajeri et al. [115] used a Schulze ring shear tester in combination with a multi-objective optimisation procedure to calibrate the parameter

values (multi-variable) for wet (cohesive) coal. The EEPA contact model was used, and due to the relatively large number of parameters associated with this model, several parameter values were taken from the literature or reasonable values were assumed. Spherical particles were used, and their rolling was restricted (i.e., rotation of the particles was not allowed). The parameters included in the calibration process were the particle-particle sliding friction coefficient, the constant pull-off force $F_0$, the surface energy $\gamma^*$ and the tensile exponent $\chi$ (Section 3.3). Contact between the particles and the wall was assumed to be non-cohesive, and the Hertz-Mindlin model was used. The shear resistance of the material was measured in the experiment under different combinations of pre-shear and shear normal pressures. This resulted in eight target values, which the optimiser tried to match. The optimised (calibrated) parameter set was verified by modelling ring shear tests with the different conditions (pre-shear consolidation pressures) used during calibration.

In the study above [115], only the ring shear tester was used to calibrate a DEM model. However, in a follow-up study by Mohajeri et al. [116], a combination of the ring shear test, an angle of repose test, and a consolidation-penetration test was used to calibrate the parameters for cohesive iron ore (8.7% moisture content). The angle of repose was measured in a so-called *ledge* test or *shear box* test (which should not be confused with the translational direct shear box). The setup consists of a rectangular box, which is filled with material, and one side of the box is rapidly removed, allowing the material to freely flow out, see Figure 14. The material that remains behind on the *ledge* forms an angle with the horizontal, which is defined as the angle of repose. This angle is also often referred to as the *shear* angle, similar to the draw-down test where the flow mechanism that creates the angle is identical (Section 4.1). Other researchers have also used this test for the calibration of cohesive iron ore (18.5% moisture content) and non-cohesive materials [89,96,117].

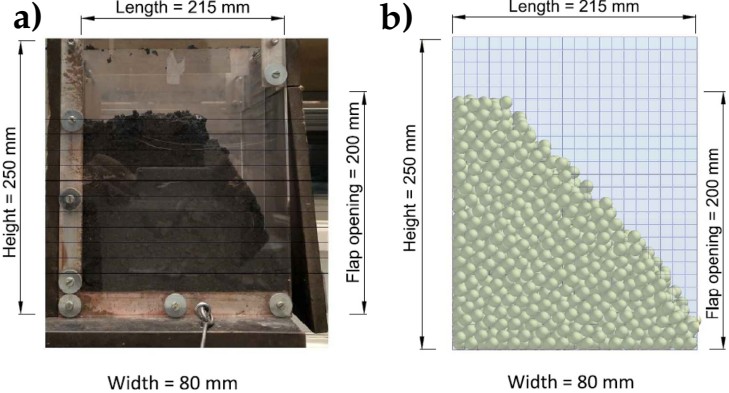

**Figure 14.** The ledge test for measuring the static angle of repose by [116] (**a**) the experiment and (**b**) the DEM model. Reproduced with permission from Advanced Powder Technology; published by Society of Powder Technology Japan, 2021.

The consolidation and penetration tests were combined, as shown in Figure 15. To this end, the material was first consolidated (compressed) using a lid. Hereafter, a penetrometer was pushed into the material at a constant velocity, and the load was measured. This provided two bulk measures which were included in the calibration procedure, namely the bulk density after consolidation with the lid and the penetration energy required to push the penetrometer a certain distance into the material.

The ring shear tests with a normal confining pressure of 20 kPa during pre-shear, followed by a confining pressure of 2 kPa during shear, allowed for the calibration of the material's history-dependent behaviour. The ledge test allowed for the calibration of the free flow of the material under no consolidation pressure, and the penetration test was representative of the interaction with equipment, such as a grab in this case. Combining a selection of these bulk measures into a single calibration process provided a comprehensive set of data for the material behaviour under different conditions.

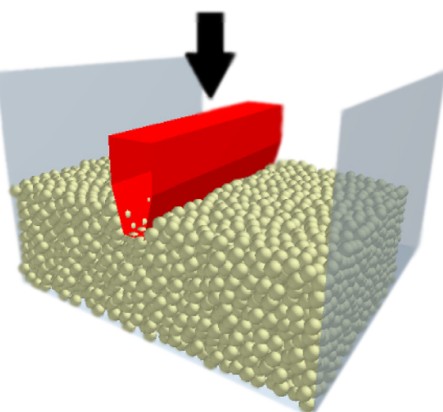

**Figure 15.** The consolidation and penetration test setup—adapted from [116]. Reproduced with permission from Advanced Powder Technology; published by Society of Powder Technology Japan, 2021.

In the first part of this study, the spherical particles were allowed to rotate, and a rolling resistance model (specific model not specified) was used in combination with the EEPA model. Under these conditions, the particle-particle coefficients of sliding and rolling friction, the shear modulus (stiffness), the constant pull-off force $F_0$, the surface energy $\gamma^*$ and the plasticity ratio $\lambda_p$ were calibrated, while all other parameter values were taken from literature or assumed. In the second part of the study, the particles were not allowed to rotate (rotation restricted), eliminating the need to calibrate the coefficient of rolling friction from the list above. It was also shown that the EEPA model performed better than the SJKR-E model in predicting the consolidation behaviour of iron ore [118].

An initial sampling strategy was used where bulk measures were selected to identify feasible parameter values. A surrogate-modelling-based optimisation process was then used to identify a unique (calibrated) set of parameter values. For the ring shear and ledge angle of repose tests, the particle-particle coefficient of friction was the most significant parameter, followed by the shear modulus (stiffness), surface energy and plasticity ratio. These results were then used to optimise a full-scale industrial grab design employed for unloading iron ore cargo from bulk carrier ships [119,120].

For other uses of the penetration test, see Aikins et al. [121] and Nalawade et al. [122], who performed a field cone penetration test to measure the stiffness of soil. The test was repeated in DEM, and the contact stiffness was adjusted until the measured bulk stiffness matched (also see Janda and Ooi [123]). For the use of ring shear tests with cohesive materials, see [28,124–126] and Pachón-morales et al. [127] for the combined use of the ledge and ring shear test.

*4.8. Methodology 8: Calibrating Highly Polydispersed Material*

With this approach, crushed copper ore was calibrated by Grima et al. [128]. Due to the fine particle and moisture content, the material was highly cohesive. In an attempt to include the effects of the fines content, the modelled material comprised two independently calibrated sub-materials: fine sub-material and coarse sub-material. After calibrating each sub-material, they were blended to represent the physical material and used to analyse the flow through a rock-box type transfer chute. Each of the sub-materials was also modelled as either spherical particles or multi-sphere (shaped) particles. The multi-sphere particles consisted of 3 to 8 sub-spheres.

The size distribution of the real material was not given, but the modelled fine particles ranged from approximately 20 mm to 42 mm in size for the spheres and from 25 mm to 50 mm for the multi-spheres. The modelled coarse particles ranged from 42 mm to 220 mm for the spheres and from 50 mm to 220 mm for the multi-spheres. Although the real particles included fines below 20 mm to 25 mm in size, these were not included in the model, and their influence, especially the cohesive effect, was accounted for in the calibration process.

The JKR model available in EDEM [50] was used. However, it was not specified if this was the full JKR model (Hertz-Mindlin with JKR Version 2) or the SJKR-A model (Hertz-Mindlin with JKR).

A large-scale ($0.2\,\mathrm{m}^3$) ledge test (described in Section 4.7 and referred to as a rock-box in this study) was used to calibrate the fine sub-material. The setup consisted of two boxes; the upper flat-bottomed box was elevated 2 m above the lower box. The material was loaded into the upper box, where the bulk density was measured, and then one wall of the box was removed, which allowed the material to drain over the edge into the lower box. The shear angle (called the *drained* angle of repose in this study) was measured in the upper box for the remaining material, Figure 16a, and the *dynamic* angle of repose was measured in the lower box where the drained material formed a pile, Figure 16b. For a similar type of test, see Grima et al. [129].

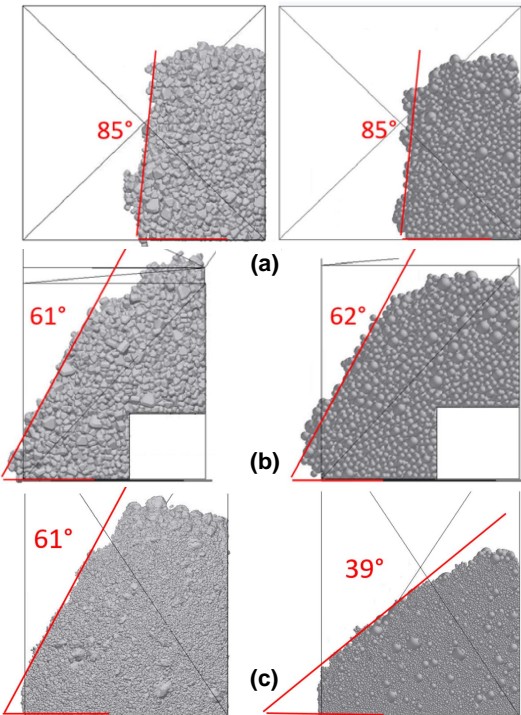

**Figure 16.** Discharge from a ledge test (rock box) as adapted from [128], showing (**a**) the shear (drained) angle in the upper box, (**b**) the dynamic angle of repose in the lower box for the fine submaterial, and (**c**) the dynamic angle of repose of the blended fine-coarse material. In all three cases, the multi-sphere particles are shown on the left and the spherical particles on the right. Reproduced with permission from A. Grima.

The shear angle was measured as 84° to 87° and the dynamic angle as 60° to 65°. These two angles could be closely matched in the model for both the spherical and multi-sphere particles by varying the parameter values (coefficient of restitution, sliding and rolling friction and surface energy), see Figure 16. As expected, the spherical particles required higher sliding and rolling friction coefficients than the multi-sphere particles. It was also observed that the spherical particles had a more 'glutinous' or viscous flow behaviour which was not observed during the experiment. The slope was also more unstable, collapsing easily, and in general, it took more effort to calibrate the spherical particles accurately.

The coarse sub-material was calibrated similarly but only using the shear angle. The two sub-materials were blended using the ratio 70/30 (fine/coarse). The shear angle of the blended material was not experimentally measured, but a simulation was performed, as shown in Figure 16c. To calibrate the interaction parameter values between the two sub-materials; it was assumed that the shear angle was equal to the internal friction angle of the copper ore at low consolidation pressure ($<20\,\mathrm{kPa}$), namely 60° to 65°. The

parameter values of the multi-sphere particles could be calibrated, but the spherical particles failed to produce a stable slope, Figure 16c. This was partially attributed to the type A [34] rolling resistance model that was used, which is known to cause material creep due to oscillations in the rolling torque. The type C model is considered the best alternative [35].

Grima et al. [128] concluded that care is required when modelling cohesive materials with spherical particles since it can result in undesirable flow behaviour. That is, most natural materials have non-spherical particle shapes that interlock and result in random trajectories, which are impossible to model realistically using spherical particles.

*4.9. Methodology 9: Calibrating Non-Cohesive and Cohesive Parameters Independently*

Grima et al. [64,65] developed and validated a systematic strategy for calibrating the parameters of dry and wet black coal (fine and coarse, respectively) and bauxite. Using the Hertz-Mindlin (no-slip) contact model, the material was first calibrated in a dry non-cohesive state based on flow property measurements and bench scale tests. Thereafter, the material was tested in a moist cohesive state, and the appropriate cohesive parameters were introduced to the DEM contact model. The cohesive parameters were calibrated, while the non-cohesive parameter values remained unchanged. Two cohesion models were investigated, SJKR-A and SJKR-D (Section 3.5), as implemented in EDEM [50] (versions 2.1.1 to 2.3.1), where they were at the time referred to as the 'Hertz-Mindlin with JKR' and 'Linear Cohesion' models, respectively.

The modelled particle size distribution (PSD) was truncated for both materials to eliminate the small particles and reduce computation time. In some cases, the remaining PSD was also scaled to increase the particle size further. The solid particle density was measured using a gas pycnometer and the water displacement method. This value was used as direct input to the model. The bulk density was measured by loosely pouring material into a metering cylinder. The particle stiffness, or Young's modulus, was taken as an approximate value based on literature (coal) or measured in a nano-indentation test (bauxite). Poisson's ratio was taken from the literature. The coefficients of restitution were measured for particle-particle and particle-wall impact in drop tests and directly used as input. The dry particle-wall sliding friction was measured in direct shear tests (Jenike tests) and an inclined wall test and used as direct input. These values were not adjusted when moisture was modelled (i.e., any lubricating effects were ignored).

The swing-arm slump test with a split cylinder was then developed to measure the angle of repose $\theta_R$ and the height of the pile $h_p$, as shown in Figure 17. However, with the traditional lifting cylinder test (Section 4.2), it has been shown that the speed at which the cylinder is lifted can influence the pile formation and hence the measured angle of repose [12]. Therefore, any uncertainty in the particle-wall interface properties, which should be accounted for in the model, might introduce further inaccuracies.

However, in the swing-arm split test, the two halves of the containing cylinder are removed very fast, significantly reducing any wall effects on the formation of the pile [65]. The swing-arm test was first used to calibrate the dry particle-particle coefficients of sliding and rolling friction through a parameter sensitivity study. Before the swing arm was opened, the modelled bulk density in the filled cylinder was compared to the measured value. Based on this density value and the height and angle of the formed heap, the most accurate parameter set could be selected for the dry material.

The second test used a flat-bottomed hopper, as shown in Figure 18 (similar to a draw-down test, Section 4.1). A parameter sensitivity study was conducted, and a parameter set that produced the most accurate results in terms of the height $h_p$ and angle $\theta_R$ of the formed heap and the angle $\theta_{R,d}$ of the remaining material, was identified. The discharge rate was estimated from high-speed photography and compared to the modelled flow. Due to the upscaled particles of the fine coal, the modelled flow rate was significantly less than the measured flow, but the resulting heap angles and height could still be compared. However, the modelled flow rate of the unscaled particles (coarse coal) compared favourably.

The force on the horizontal impact plate below the hopper was also measured and compared to the model results to validate the selected parameter set.

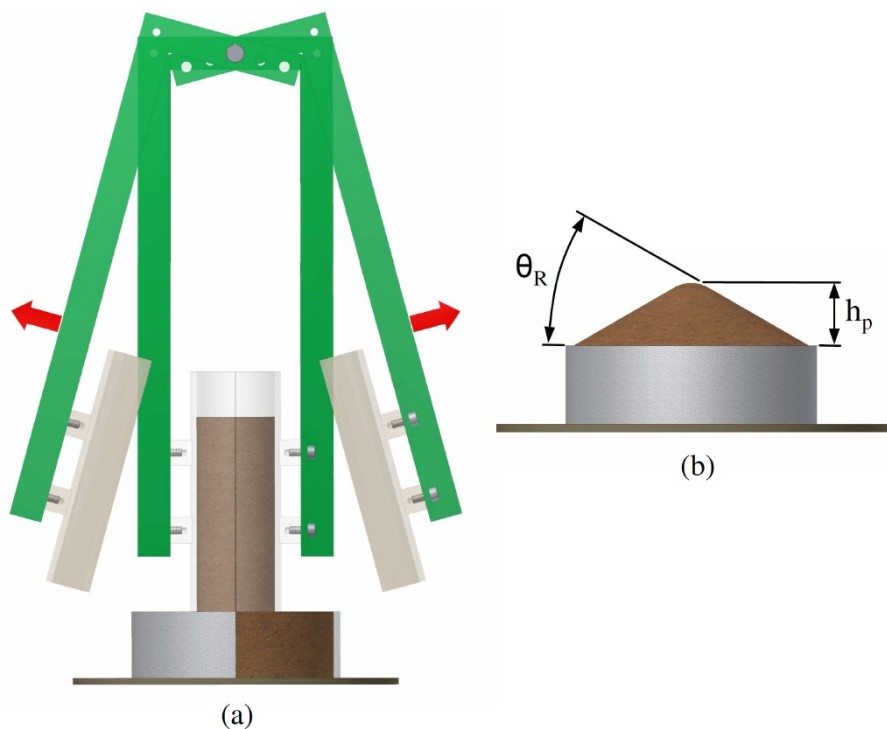

**Figure 17.** Angle of repose slump tester introduced by [64] showing (**a**) the initial configuration and opening mechanism, and (**b**) the formed heap. Reproduced with permission from A. Grima.

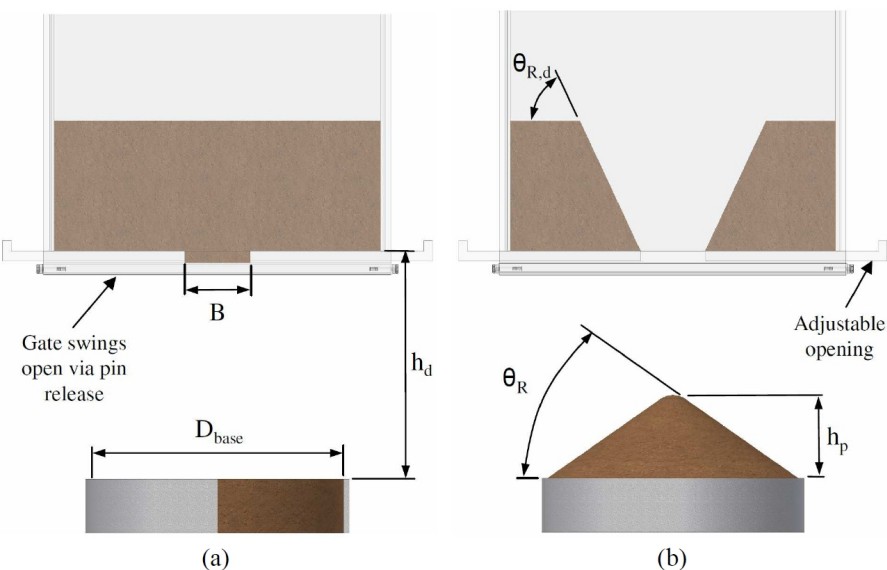

**Figure 18.** Discharge from a flat-bottomed hopper by [64], showing (**a**) the initial configuration with all of the material in the hopper, and (**b**) the discharged configuration with the heap at the bottom forming and remaining material in the hopper. Reproduced with permission from A. Grima.

The particle-particle sliding friction was identical in both the slump and hopper discharge tests, but the hopper discharge resulted in a lower value for the rolling friction than the slump test. It raises some concern that a single set of parameter values, accurately modelling both tests, could not be identified for the dry material.

With the parameters of the non-cohesive (dry) material calibrated, the cohesion energy density, $C_0$, was estimated based on the results from the Ajax tensile test. The swing-arm slump test was then repeated for the moist cohesive material. Compared to the dry material, the shape of the pile was more irregular, which made it difficult to define the angle of repose. For this reason, the whole shape of the heap was analysed and compared to the model results. The value of the cohesion energy density was adjusted until a good correlation was obtained. The process was repeated with the flat-bottomed hopper, and it was found that due to the slumping action of the material as it hit the lower boundary, the cohesion energy density had no significant effect on the angle of repose $\theta_R$ that formed. However, it had a significant effect on the shear angle $\theta_{R,d}$ of the material remaining behind in the hopper. Hence, only $\theta_{R,d}$ was used to calibrate the cohesion energy density. Finally, comparing the calibrated cohesion energy density from the two tests showed a significant difference in the calibrated values.

The model also failed to accurately model the bulk density, which was attributed to the limitations of the SJKR-D model, which could not predict the higher porosity when moisture was added since the cohesive force is zero when the particle overlap is zero. The slump tests were filled to different heights for the given cylinder diameters. This resulted in the total system energy being slightly different, affecting the model's contact overlap. Since the magnitude of the cohesive force is determined by the contact area/overlap, the cohesive forces were slightly larger in magnitude for a system with more energy. This highlights the importance of carefully selecting the contact stiffness. A reduced contact stiffness will not only influence the bulk stiffness/compressibility as in non-cohesive materials but also the bulk cohesion. Grima and Wypych [65] concluded that a calibrated cohesion parameter for one system might not be suitable for another system. This might also be one of the reasons why the slump and hopper tests resulted in different calibrated cohesive parameters.

Furthermore, it was concluded that the SJKR-A contact model was superior to the SJKR-D contact model in modelling particle cohesion and clumping (agglomeration) and retaining a stable static pile. However, there was no significant difference in the particle velocities and flow patterns on a bulk level through a large-scale transfer chute. The results from the SJKR-D model were sufficient for simulating chute flow and computationally less expensive than the SJKR-A model.

### 4.10. Methodology 10: Calibrating Non-Cohesive and Cohesive Parameters Independently

In a recent investigation undertaken by the authors of this paper [130,131], a detailed investigation into the calibration of cohesive materials was performed. This resulted in a systematic approach, similar to the study in Section 4.9, where the parameters for the non-cohesive material were first calibrated, after which the cohesive parameters were introduced and calibrated.

Sand with three distinct grades (particle size distributions, PSD), namely fine, medium and coarse, was used, and the moisture content varied (up to 15% saturation). A series of experiments were performed to measure the bulk response of the material for each grade and moisture content combination.

The experiments were then modelled using multi-sphere particles and a range of PSDs. The coarse sand particles could be modelled at a scale of 1:1, but the medium and fine particles had to be upscaled to reduce computation time. The linear cohesive model was used, as described in Section 3.7. Here, the aim was to identify the best combination of experiments needed to calibrate a unique set of parameter values for each sand grade, moisture content, and particle scale factor combination.

The bulk stiffness was measured in a confined uniaxial test, which was used to calibrate the contact stiffness. However, to reduce computation time, the stiffness was reduced by a factor of up to 100 without any significant effect on the results. The stiffness was also independent of the moisture content within the range tested. A large ring shear tester showed that the experimental and numerical results were not sensitive to changes in the

moisture content but could be used to calibrate the coefficient of sliding friction (rolling resistance not included due to the shaped particles), which was then kept constant and the two cohesion parameters calibrated using the experiments as described below.

The static angle of repose was measured in a setup where a cylinder was filled with material, and a circular internal plate was displaced vertically, pushing through the material, and creating a pile on top of the plate. The resulting angle of repose is shown in Figure 19 for the three sand grades.

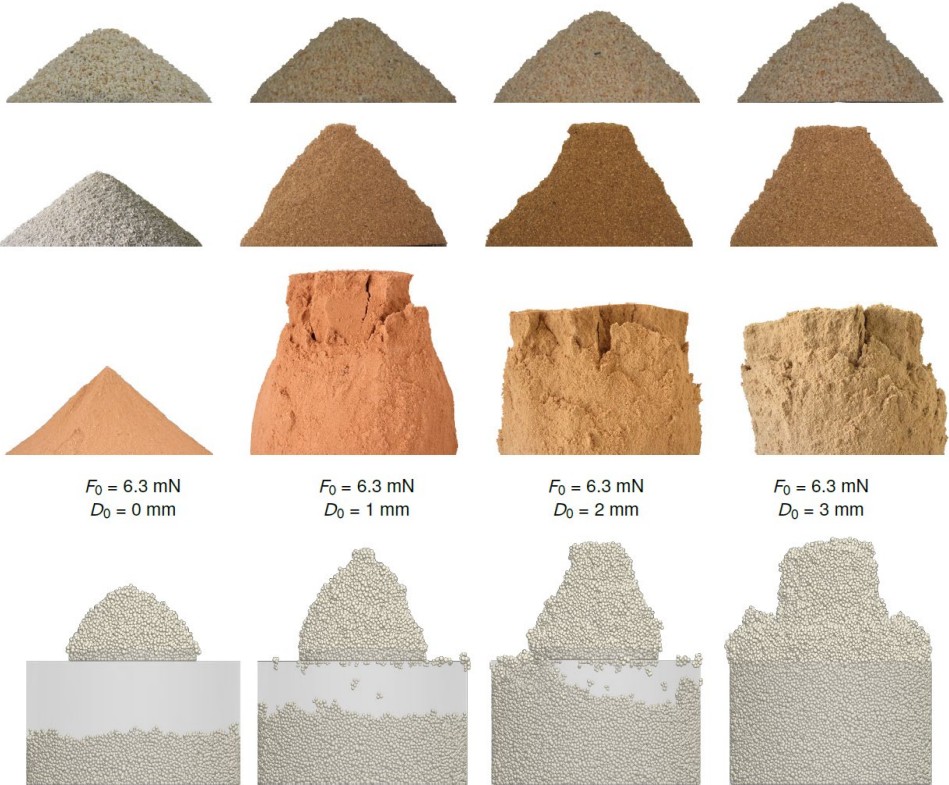

**Figure 19.** The angle of repose in the vertical displacement setup, from top to bottom: the coarse sand, medium sand, fine sand, and model for the medium sand, and from left to right: dry material, 5%, 10% and 15% saturation. The linear cohesive model was used with the values of the two cohesion parameters, $F_0$ and $D_0$, indicated for each saturation level.

A draw-down test was conducted, similar to those discussed in Section 4.1. It was found that the shear angle (upper box) and the angle of repose (lower box) increased with an increase in moisture content. However, for the fine grade and higher moisture contents, the shear angle was very steep, approaching zero, and somewhat insensitive to a further increase in the moisture content.

A rotating drum was used to measure the continuous flow behaviour of the material. A clear, dynamic angle of repose could be identified for the material in its dry state, similar to that in other studies where non-cohesive and slightly cohesive materials were used [8,32,132]. However, with the introduction of moisture, the behaviour of the material changed significantly, and it was no longer possible to define the angle of repose. However, it was shown that by tracking the centroid of the material bed (using image processing), the height of the centroid above the lower edge of the drum (or the centroid angle defined as $\psi$ in Figure 20) provided a good measure of the material's bulk cohesiveness. With an increase in the moisture content, the steady state height of the centroid increased. See Faqih et al. [133] and Thakur et al. [134] for a similar approach.

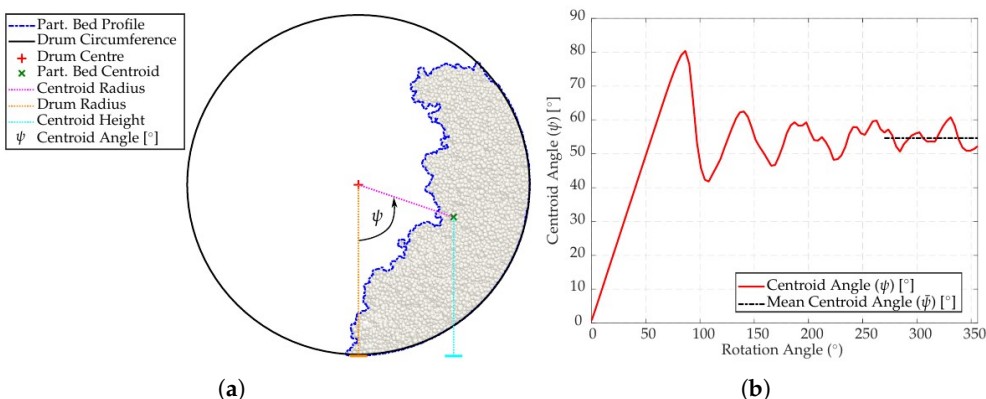

**Figure 20.** (**a**) The measurement of the material bed's centroid (Particle bed analysis) and (**b**) the centroid angle $\psi$ as a function of time (Mean centroid angle).

Due to the steep angles created by the fine sand and higher moisture content in the angle of repose (vertical displacement method) and draw-down tests, a new experiment was proposed to induce material flow and bulk failure until an angle is formed. For this purpose, a centrifuge was designed with a small rectangular bucket attached. The bucket was filled with material, and the free surface levelled, after which the centrifuge was accelerated incrementally until the lateral acceleration reached a value of ten times the gravitational acceleration on earth (10 $g$). As the lateral acceleration increased, the material moved outwards and formed an angle (slope) with the horizontal, which was captured by a camera, Figure 21. Even for the fine sand with the highest moisture content level, a clear angle was formed, which could be plotted (using image processing) as a function of the lateral acceleration, Figure 22.

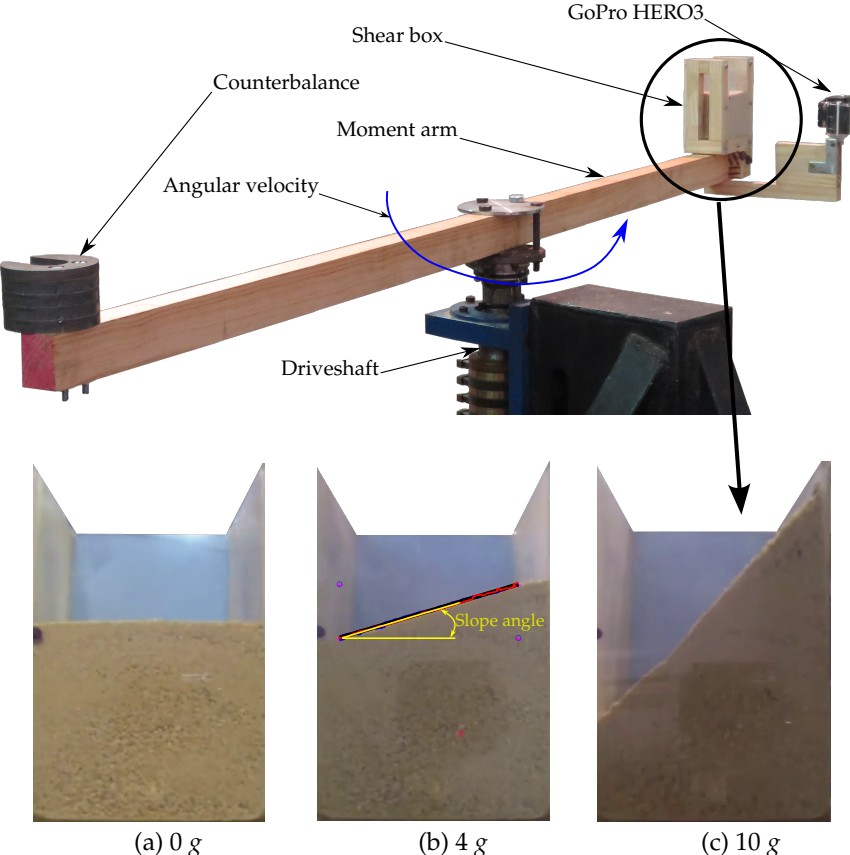

**Figure 21.** The centrifuge setup with the view from the camera shown in (**a**) for a lateral acceleration of 0 $g$, (**b**) 4 $g$ and (**c**) 10 $g$.

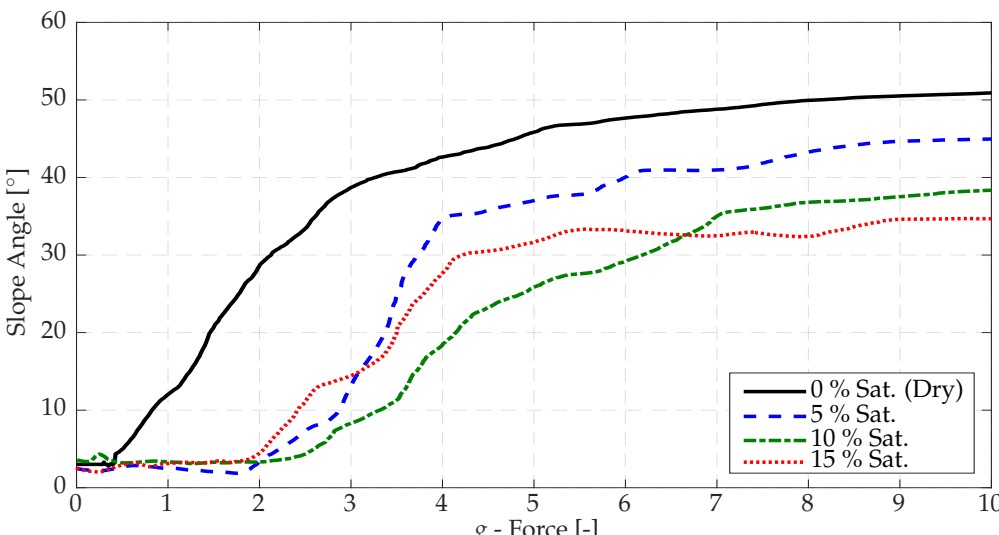

**Figure 22.** The slope of the fine sand as measured in the centrifuge test for various moisture contents and as a function of the lateral acceleration (g-force).

In all three experiments, the vertical displacement angle of repose, the rotating drum, and the centrifuge were modelled, and the cohesive parameters (pull-off force $F_0$ and pull-off distance $D_0$) systematically varied in a full-factorial parameter study. Plotting and overlaying the response surfaces (contour plots as used by [8,13,86,89]), a unique and calibrated set of parameter values could be identified for each sand grade, moisture content and particle scale factor combination.

Results also showed that for the medium and coarse sand grades, which were less cohesive in general, the combination of the angle of repose, drawdown and rotating drum tests provided the best calibration results. On the other hand, for the fine and most cohesive sand grade, the combination of the angle of repose and centrifuge tests provided the best results. Furthermore, contrary to the other tests, the centrifuge provides a large number of data points per single test. The angle that the material makes is recorded as a function of the lateral acceleration, providing a full curve (Figure 21) to use as a calibration target. As such, the contour plots from this test alone provided a close to a unique set of cohesive parameter values. Also, a small material sample is needed for this test, and the simulation time is significantly shorter than any of the other tests. Also, the behaviour is mostly influenced by particle-particle properties without the unknown particle-wall properties of an actuating or moving wall, such as in ring shear tests.

## 5. Summary

DEM modelling is computationally expensive, and users always try to find ways to decrease computation time, such as particle upscaling (reducing the total number of particles in the model), reducing the contact stiffness (increasing the stable time-step) and simplifying the particle shape as much as possible.

A small number of studies compared the performance of the different contact models. Roessler and Katterfeld [91] found that under close to identical model setups, the computation time of the EEPA model was approximately 100% (2 times) longer than that of the SJKR-E model. The linear elastic-plastic-adhesion model by Luding [43,135] was, on average, 20% slower than the SJKR-E model. Note that all of the SJKR-B to SJKR-F models are nearly identical and should not have a difference in their computational performance.

Ajmal et al. [86] compared the full JKR model to the SJKR-E model and found that the full JKR model was slower by 25% to 30%. Grima [64] showed that the SJKR-D model took 15% longer to compute than the classic Hertz-Mindlin model (without cohesion). The SJKR-A model took 150% (2.5 times) longer than the Hertz-Mindlin model to compute and 118% (2.2 times) longer than the SJKR-D model. Carr [88] showed that the EEPA

model took approximately eight times longer than his hybrid model (the liquid-bridge and SJKR-E model for particle-particle and particle-wall contact, respectively). However, he also mentioned that the implementation of the EEPA model that he used, was not optimised, which could increase computation time by approximately 20%.

From these results, it is clear that the performance of cohesive contact models is slower than that of non-cohesive models, as expected. However, the performance of the different models depends on the specific implementation and software (and hardware) used, which makes any direct and fair comparison difficult. For a comparison between the cohesive models implemented in EDEM and LIGGGHTS see Ramírez-Aragón et al. [136].

## 6. Conclusions and Outlook

A summary and detailed formulation of the most prominent DEM contact models used for the modelling of cohesive granular materials is presented. Using a consistent notation, this paper should help future users and researchers to easily compare the different models and to select the most appropriate model for a specific application. Furthermore, a new naming convention for the series of simplified Johnson-Kendall-Roberts models (SJKR) is introduced. Currently, different names are used, and the same name is often used to describe different implementations. The new naming convention provides an unambiguous definition of these models and will help users to better interpret the parameter values published by others.

The accurate calibration of any DEM model is vital. The calibration of non-cohesive materials has reached a level of maturity, but the calibration of cohesive materials still needs some research. The calibration of non-cohesive materials usually includes measuring at least one angle, such as a shear angle or angle of repose, representative of the bulk behaviour. However, using the same experiments with cohesive materials, the angles are inconsistently reproduced or steep and insensitive to further increases in cohesion.

A critical review of the state-of-the-art experiments and calibration procedures for cohesive materials is presented based on the literature. In most cases, this still uses experiments where the material is allowed to gravity-flow from a container, and various angles are measured after the material comes to rest. The dynamic angle of repose, measured in a rotating drum, also produces inconsistent results, but other measures, such as the time evolution of the bed's free surface or the position of the centroid, can be tracked. However, calibration experiments should ideally be easy and quick to perform, especially if a large range of materials or material conditions (moisture content) need to be analysed. Sophisticated analysis, such as complex image processing and tracking, can render a specific experiment less attractive for everyday use.

To overcome the problem of steep angles formed by cohesive materials and experiments that are insensitive to changes in cohesion, the use of a centrifuge might be a solution [130,131]. The centrifuge-induced acceleration can be increased until the material flows (fails). This results not only in an angle (or angles) that can be measured, but the complete acceleration-angle history is available as a calibration target. However, further research is needed to assess the accuracy of this methodology for calibrating a wide range of materials under various conditions and flow mechanisms.

**Author Contributions:** Conceptualization, C.J.C.; methodology, C.J.C. and O.C.S.; writing—original draft preparation, C.J.C. and O.C.S.; writing—review and editing, C.J.C. and O.C.S.; supervision, C.J.C.; funding acquisition, C.J.C. All authors have read and agreed to the published version of the manuscript.

**Funding:** This research was funded by the National Research Foundation (NRF), grant number 137952.

**Institutional Review Board Statement:** Not applicable.

**Informed Consent Statement:** Not applicable.

**Data Availability Statement:** Not applicable.

**Conflicts of Interest:** The authors declare no conflict of interest.

## Nomenclature

**Acronyms and Abbreviations**

| | |
|---|---|
| AOR | Angle of Repose |
| DEM | Discrete Element Method/Model |
| EEPA | Edinburgh Elasto-Plastic Adhesion Model |
| JKR | Johnson, Kendall & Roberts |
| Part. | Particle |
| PSD | Particle Size Distribution |
| Sat. | Saturation |
| SJKR | Simplified Johnson, Kendall & Roberts |

**Roman Symbols**

| | |
|---|---|
| $A_c$ | Cohesive contact area [m$^2$] |
| $a$ | Radius of the circular contact area [m] |
| $a_0$ | Radius of the contact area at zero external force [m] |
| $a_{po}$ | Contact radius at the JKR's maximum tensile force [m] |
| $a_{to}$ | Contact radius at the JKR's rupture tensile force [m] |
| $B_1$ | Initial hopper opening width—§ 4.5 [m] |
| $B_2$ | Finial hopper opening width—§ 4.5 [m] |
| $C_0$ | Cohesion energy density [J/m$^3$] |
| $D_0$ | Liquid-bridge rupture distance [m] |
| $d$ | Distance between particle centres [m] |
| $E$ | Young's/Elastic modulus [Pa] |
| $E_{1,2}$ | Young's/Elastic modulus of the contacting bodies [Pa] |
| $E^*$ | Effective Young's/Elastic modulus [Pa] |
| $F_0$ | Normal (rupture) force at zero displacement/overlap [N] |
| $F_{min}$ | Maximum tensile hysteretic force [N] |
| $F_n$ | Normal contact force component [N] |
| $F_n^a$ | Adhesive component of the normal force [N] |
| $F_n^{EEPA}$ | EEPA contact force [N] |
| $F_n^H$ | Hertzian elastic component of the normal force [N] |
| $F_n^{JKR}$ | JKR adhesive contact force [N] |
| $F_n^L$ | Luding's model contact force [N] |
| $F_n^{LB}$ | Liquid-bridge contact force [N] |
| $F_n^{LC}$ | Linear cohesive contact force [N] |
| $F_n^{SJKR}$ | Generalised SJKR normal force formulation [N] |
| $F_{po}$ | Maximum tensile force from molecular attraction [N] |
| $F_s$ | Shear contact force [N] |
| $\boldsymbol{F_s^*}$ | Trial shear contact force [N] |
| $\boldsymbol{F_s^t}$ | Shear force at time-step start [N] |
| $\boldsymbol{F_s^{t+1}}$ | Updated shear force [N] |
| $F_{to}$ | JKR tensile force at rupture [N] |
| $g$ | Gravitational acceleration [m/s$^2$] |
| $h$ | Separation distance [m] |
| $h_p$ | Swing-arm pile height—§ 4.9 [m] |
| $k_1$ | Effective normal elastic initial loading stiffness [N/m] |
| $k_2$ | Effective normal elastic unloading/reloading stiffness [N/m] |
| $k_2^{max}$ | Maximum allowable unloading/reloading stiffness [N/m] |
| $k_a$ | Effective adhesive normal elastic unloading stiffness [N/m] |
| $k_{af}$ | Adhesion stiffness factor [-] |
| $k_n$ | Effective normal elastic stiffness [N/m] |
| $k_r$ | Rolling resistance contact stiffness [N · m] |
| $k_s$ | Effective shear contact stiffness [N/m] |
| $\boldsymbol{M_r}$ | Rolling resistance moment [N · m] |
| $\boldsymbol{M_r^*}$ | Trial rolling resistance moment [N · m] |

**Roman Symbols**

| | |
|---|---|
| $M_r^t$ | Rolling resistance moment at time-step start [N · m] |
| $M_r^{t+1}$ | Updated rolling resistance moment [N · m] |
| $m$ | Loading/unloading/reloading exponent [-] |
| $m_{1,2}$ | Masses adhering to the plates—§ 4.3 [kg] |
| $R$ | Spherical radius [m] |
| $R_{1,2}$ | Radii of the contacting bodies [m] |
| $R_h$ | Harmonic mean radius [m] |
| $R_{min}$ | Minimum radius of the contacting pieces [m] |
| $R_p$ | Particle radius [m] |
| $R^*$ | Effective radius of curvature [m] |
| $r_1$ | Radius of curvature of the air-liquid interface [m] |
| $r_2$ | Radius of the cross-sectional area of the liquid-bridge [m] |
| $S$ | Liquid saturation [-] |
| $t$ | Incremental time-step [s] |
| $V_L$ | Liquid volume [m$^3$] |
| $V_{LB}$ | Liquid-bridge volume [m$^3$] |
| $V_V$ | Void volume [m$^3$] |

**Greek Symbols**

| | |
|---|---|
| $\alpha$ | Hopper angle—§ 4.5 [°] |
| $\alpha$ | Upper shear angle—§ 4.3 [°] |
| $\beta$ | Half-filling angle [°] |
| $\beta$ | Lower shear angle—§ 4.3 [°] |
| $\gamma$ | Surface tension [N/m] |
| $\gamma_{1,2}$ | Intrinsic surface energies of the contacting bodies [J/m$^2$] |
| $\gamma_{12}$ | Interface energy [J/m$^2$] |
| $\gamma^*$ | Effective surface tension [N/m] |
| $\gamma^*$ | Work of adhesion/adhesion (surface) energy density [J/m$^2$] |
| $\Delta\delta_s$ | Shear displacement/overlap increment [m] |
| $\Delta\theta_b$ | Relative bend-rotation increment [rad] |
| $\delta$ | Displacement/Overlap [m] |
| $\delta_d$ | Separation distance [m] |
| $\delta_{ip}$ | Particle-particle immersion depth [m] |
| $\delta_{iw}$ | Particle-wall immersion depth [m] |
| $\delta_{max}$ | Maximum adhesive hysteretic displacement/overlap [m] |
| $\delta_{max}^*$ | Plastic overlap limit [m] |
| $\delta_{min}$ | Displacement at maximum tensile hysteretic force [m] |
| $\delta_n$ | Normal displacement/overlap [m] |
| $\dot{\delta}_n$ | Normal deformation rate [m/s] |
| $\delta_o$ | Liquid-bridge minimum separation distance [m] |
| $\delta_p$ | Adhesive hysteretic plastic displacement/overlap [m] |
| $\delta_{to}$ | JKR's negative displacement/overlap at rupture [m] |
| $\eta_n$ | Effective normal damping constant [kg/s] |
| $\theta$ | Contact/Wetting angle [°] |
| $\theta_{1,2}$ | Collection tray angles (AOR)—§ 4.3 [°] |
| $\theta_R$ | Swing-arm slump tester AOR—§ 4.9 [°] |
| $\theta_{R,d}$ | Flat-bottomed hopper residual angle—§ 4.9 [°] |
| $\lambda_p$ | Contact plasticity ratio [-] |
| $\mu$ | Coulomb-type friction coefficient [-] |
| $\mu_r$ | Rolling friction coefficient [-] |
| $\nu_{1,2}$ | Poisson's ratios of the contacting bodies [-] |
| $\Phi_{AOR}$ | Draw down angle of repose—§ 4.1 [°] |
| $\Phi_L$ | Liquid volume fraction [-] |
| $\Phi_S$ | Solid volume fraction [-] |
| $\Phi_{shear}$ | Draw down shear angle—§ 4.1 [°] |
| $\phi_f$ | Dimensionless plasticity depth [-] |
| $\chi$ | EEPA adhesive branch deformation exponent [-] |
| $\psi$ | Centroid angle—§ 4.10 [°] |
| $\bar{\psi}$ | Mean centroid angle—§ 4.10 [°] |

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
