# Peer review of "Review: The Calibration of DEM Parameters for the Bulk Modelling of Cohesive Materials"

_processes, doi:10.3390/pr11010005_

Round 1

Reviewer 1 Report

Please check the attached file

Author Response

Dear reviewer, please see the attached document for feedback.

Reviewer 2 Report

The authors present in this article a review of the different possible integrations of cohesive interaction inside some classical DEM codes. We can distinguish mainly two big parts in their article; Firstly, the state of the art of JKR (Johnson, Kendall and Roberts model) different formulations and secondly, the uses of these formulations through different Open-source codes and their results obtained in some particular cases in mining industries.

In an article of 90 pages, we can expect obtaining very interesting and useful information for future use of the readers, but, in somehow, I was disappointed by the poor quality of the part 4 (comparison of the use). Indeed, the part 2 is just a very basic review of the well-known list of cohesive forces and the classical schematic draw in figure 1. Even here, the use a density of 1000kg/m^3 for comparison is quite small for ore materials!

The part 3 is really the most important one(400 lines) and, in fact, the only useful one. The clear and complete description of the different contact models (JKR, Luding, EEPA and SJKR) and the classification from SJKR-A to F is nicely done and can be a good introduction for new researcher involved in the integration or the use of cohesive term in DEM codes. The §3.5.7 which explains which implementation is done by default in different Open-source or commercial codes is a very interesting and useful information.

By opposition, the part 4 is very less interesting and productive for readers which want to use it for their future works! 500 lines to only present general details of several uses (10 study cases!) of the different models is not what we can expect for a review. The descriptions of the setup-behavior of the model of  the numerical simulations are needed but are not enough to understand the quality of the results made by these uses. The main goal of this article would be to allow the readers to extract the good model for their future use which is not possible here: no practical results, qualities, or limitation of the model are clearly presented for each study cases. For the description of the ten studies, it is necessary to make clear description of the system as 2Dor 3D cases!

So, in conclusion, the paper would gain in quality if the part 4 would be more accurate in the results analysis: the description of the 10 study cases is interesting only if the observation of the data results is improved.

Detail:

As it is a very large article with also a long list of symbols and term definitions, a nomenclature list could help the reader all along this article.

Maybe also a table to summarize the difference between all the presented models could be useful.

For the first use of PSD and other abbreviation it could be nice to write the full term (PSD definition is written in line1063 instead of line987 : 1st use)

Line 944 meaning of 200L ???

Figure 19 the description of the real experiments and the simulated ones are not enough clear: what is the meaning of F0/ D0 in the middle of the 4 by 5 different figures? Why two rows of simulated cases as the caption mentions medium sand case only?

Author Response

(The authors gave the same response as above.)

Reviewer 3 Report

The material of the article is very interesting and well structured. It is clear that this is an analytical review of known studies. But at the same time, the personal views of the authors on the topic presented in the article are clearly justified.

Personally, for me and my colleagues from my institute, this article aroused significant interest. The issue of modeling the behavior of loose cohesive fine fraction materials is quite complex. And despite the presence of a large number of studies in this direction, this area has a significant number of unsolved problems.

In addition, software implementation of DEM models is very difficult. But not all software tools are able to accommodate the needs of researchers. Therefore, the relevance of this publication is unconditional. It allows you to organize the ideas about the existing models of cohesive materials.

A detailed critical analysis of these models is made. This is very useful information for researchers. I definitely recommend it for publication.

As a remark, I will note the following. Authors should carefully review the text. All notations in the formulas must be deciphered in the text.

Author Response

(The authors gave the same response as above.)
